# Scalable Feature Learning on Huge Knowledge Graphs for Downstream Machine Learning

**Félix Lefebvre**
SODA Team, Inria Saclay
`felix.lefebvre@inria.fr`

**Gaël Varoquaux**
SODA Team, Inria Saclay
Probabl

## Abstract

Many machine learning tasks can benefit from external knowledge. Large knowledge graphs store such knowledge, and embedding methods can be used to distill it into ready-to-use vector representations for downstream applications. For this purpose, current models have however two limitations: they are primarily optimized for link prediction, via local contrastive learning, and their application to the largest graphs requires significant engineering effort due to GPU memory limits. To address these, we introduce SEPAL: a Scalable Embedding Propagation ALgorithm for large knowledge graphs designed to produce high-quality embeddings for downstream tasks at scale. The key idea of SEPAL is to ensure global embedding consistency by optimizing embeddings only on a small core of entities, and then propagating them to the rest of the graph with message passing. We evaluate SEPAL on 7 large-scale knowledge graphs and 46 downstream machine learning tasks. Our results show that SEPAL significantly outperforms previous methods on downstream tasks. In addition, SEPAL scales up its base embedding model, enabling fitting huge knowledge graphs on commodity hardware. Our code is available at: https://github.com/soda-inria/sepal.

## 1 Introduction: embedding knowledge for downstream tasks

**External knowledge for machine learning** Bringing general knowledge to a machine-learning task revives an old promise of making it easier via this knowledge [Lenat and Feigenbaum, 2000]. Indeed, data science is often about entities of the world –persons, places, organizations– that are well characterized in general-purpose knowledge graphs. These graphs carry rich information, including numerical attributes and relationships between entities, and can be connected to string values in tabular data through entity linking techniques [Mendes et al., 2011, Foppiano and Romary, 2020, Delpeuch, 2019]. A thorny challenge, however, is to transform this relational information into features for downstream tabular machine learning [Kanter and Veeramachaneni, 2015, Cappuzzo et al., 2025, Robinson et al., 2024]. To that end, a scalable solution is offered by graph embedding methods that distill the graph information into node features readily usable by any downstream tabular learner [Grover and Leskovec, 2016, Cvetkov-Iliev et al., 2023, Ruiz et al., 2024].

**Knowledge graphs as general knowledge sources** The rapid growth of general-purpose knowledge graphs brings the exciting prospect of a very *general* feature enrichment. Indeed, a richer knowledge graph provides more comprehensive coverage and context, thereby bringing greater value to the downstream analysis [Ruiz et al., 2024]. ConceptNet pioneered the distribution of general-knowledge embeddings, building on a graph of 8 million entities [Speer et al., 2017]. Since then, knowledge graphs have continued to expand rapidly. For instance, as of 2025, Wikidata [Vrandečić and Krötzsch, 2014] describes 115M entities and gains around 15M yearly [Wikimedia], and YAGO4 [Pellissier Tanon et al., 2020] gives a curated view on 67M entities.

39th Conference on Neural Information Processing Systems (NeurIPS 2025).

**Optimizing embeddings for the right task**    In parallel, the sophistication of knowledge-graph embedding (KGE) models is increasing [Bordes et al., 2013, Yang et al., 2015, Balazevic et al., 2019a], capturing better the relational aspect of the data, important for downstream tasks [Cvetkov-Iliev et al., 2023]. However, most of the KGE literature prioritizes link prediction as the primary benchmark, despite recent findings showing that strong performance on this task does not correlate with improved performance on downstream predictive tasks [Ruffinelli and Gemulla, 2024]. One reason may be that, for link prediction, models typically optimize for local contrasts, resulting in embeddings that are not calibrated [Tabacof and Costabello, 2020, Arakelyan et al., 2023]. While prior work has explored multi-hop reasoning to capture more complex graph patterns [Hamilton et al., 2018, Ren and Leskovec, 2020], the standard evaluation paradigm for KGEs still revolves around *internal* tasks, rather than how the learned embeddings can transfer knowledge to practical machine-learning tasks beyond the knowledge graph itself.

**The importance of scalability**    To leverage the full potential of very large knowledge graphs, embedding methods *must* be highly scalable. While many methods have been proposed to scale KGE models, doing so is not trivial. Sophisticated KGE models are typically demonstrated on small datasets like FB15k (15k entities) or WN18 (40k entities), which are orders of magnitude smaller than modern general-purpose or industrial knowledge graphs [Sullivan, 2020]. The common solution to this scalability challenge is either distributed computation across multiple GPUs or machines [Lerer et al., 2019, Zhu et al., 2019, Zheng et al., 2020, Dong et al., 2022, Zheng et al., 2024], or leveraging the full memory hierarchy (disk, CPU, and GPU) on a single machine [Mohoney et al., 2021, Ren et al., 2022]. These approaches require significant engineering effort to manage data partitioning, optimize data movement, and minimize synchronization overheads.

**Contributions**    In this paper, we aim to bridge the gap between advances in embedding methods and the goal to create large and reusable general-knowledge embeddings for downstream applications. We introduce SEPAL, a scalable algorithm that applies as a wrapper to many embedding models. Our contributions are:

1. We propose a new embedding optimization strategy that enforces global consistency among positive triples. Instead of optimizing all embeddings with local contrastive learning, SEPAL first processes a small but dense *core* of the graph, to learn relation and core-entity embeddings. It then propagates these embeddings to the remaining entities using relation-aware message passing. The absence of negative sampling at this propagation stage accelerates the embedding computation and makes them better suited for downstream tasks.

2. We provide a theoretical analysis showing that SEPAL's propagation step, combined with DistMult, implicitly maximizes the alignment of embeddings within positive triples.

3. We introduce BLOCS, a scalable graph-splitting algorithm that partitions huge, scale-free graphs into manageable, overlapping subgraphs. This enables fitting the embedding process on a single GPU, avoiding the engineering complexity of distributed systems. Here, the challenge lies in the scale-free and connectivity properties of a large knowledge graph: some nodes are connected to a significant fraction of the graph, while others are hard to reach.

4. We conduct an extensive empirical study on 7 large knowledge graphs and 46 downstream tasks. Results show that SEPAL significantly outperforms standard methods on downstream tasks and is generally faster than existing large-scale systems. Moreover, it scales to ultra-large graphs with little computational resources: we embed WikiKG90Mv2 –91M entities, 601M triples– with a single 32GB V100 GPU. Our experimental results also highlight that using such large knowledge graphs is beneficial for downstream tasks in real-world feature enrichment scenarios.

We start by reviewing related work in section 2. Then, section 3 describes our contributed method and section 4 gives a theoretical analysis. In section 5 we evaluate SEPAL's performance on knowledge graphs of increasing size between YAGO3 [2.6M entities, Mahdisoltani et al., 2014] and WikiKG90Mv2 [91M entities, Hu et al., 2020]; we study the use of the embeddings for feature enrichment on 46 downstream machine learning tasks, showing that SEPAL makes embedding methods more tractable while generating better embeddings for downstream tasks. Finally, section 6 discusses the contributions and limitations of SEPAL.

Table 1: Expression of $\phi$ in some embedding models. $\odot$ denotes the Hadamard product, $\otimes$ the Hamilton product, and $\times_i$ the tensor product along mode $i$. The models we list here are all compatible with our proposed SEPAL approach.

| Model | Relational operator $\phi$ |
|---|---|
| TransE [Bordes et al., 2013] | $\boldsymbol{\theta}_h + \boldsymbol{w}_r$ |
| MuRE [Balazevic et al., 2019a] | $\boldsymbol{\theta}_h \odot \boldsymbol{\rho}_r - \boldsymbol{w}_r$ |
| RotatE [Sun et al., 2019] | $\boldsymbol{\theta}_h \odot \boldsymbol{w}_r$ |
| QuatE [Zhang et al., 2019] | $\boldsymbol{\theta}_h \otimes \boldsymbol{w}_r$ |
| DistMult [Yang et al., 2015] | $\boldsymbol{\theta}_h \odot \boldsymbol{w}_r$ |
| ComplEx [Trouillon et al., 2016] | $\boldsymbol{\theta}_h \odot \boldsymbol{w}_r$ |
| TuckER [Balazevic et al., 2019b] | $\boldsymbol{\mathcal{W}} \times_1 \boldsymbol{\theta}_h \times_2 \boldsymbol{w}_r$ |

## 2   Related work: embedding optimization and scalability

Knowledge graphs are multi-relational graphs storing information as triples $(h, r, t)$, where $h$ is the head entity, $r$ is the relation, and $t$ is the tail entity. We denote $\mathcal{V}$ and $\mathcal{R}$ respectively the set of entities and relations, and $\mathcal{K}$ the set of triples of a given knowledge graph ($\mathcal{K} \subset \mathcal{V} \times \mathcal{R} \times \mathcal{V}$).

### 2.1   Optimizing knowledge-graph embeddings

Here, we provide an overview of approaches to generate low-dimensional (typically $d = 100$) vector representations for the entities, that can be used in downstream applications. These include both graph-embedding techniques, that leave aside the relations, and KGE methods accounting for relations.

**Graph embedding**   A first simple strategy to get very cost-effective vector representations is to compute *random projections*. This avoids relying on –potentially costly– optimization, and provides embeddings preserving the pairwise distances to within an epsilon [Dasgupta and Gupta, 2003]. FastRP [Chen et al., 2019a] proposes a scalable approach, with a few well-chosen very sparse random projections of the normalized adjacency matrix and its powers.

Another family of methods performs explicit *matrix factorizations* on matrices derived from the adjacency matrix, for instance GraREP [Cao et al., 2015] or NetMF [Qiu et al., 2018]. These methods output close embedding vectors for nodes with similar neighborhoods.

Similarly, *Skip-Gram Negative Sampling* (SGNS), behind word2vec [Mikolov et al., 2013a,b], performs an implicit factorization [Levy and Goldberg, 2014]. It has been adapted to graphs: DeepWalk [Perozzi et al., 2014] and node2vec [Grover and Leskovec, 2016] use random walks on the graph to generate "sentences" fed to word2vec. LINE [Tang et al., 2015] explores a similar strategy varying edge sampling. Here, the loss function is typically a binary logistic regression objective:

$$\mathcal{L}_{\text{SGNS}} = -\log \sigma(\boldsymbol{\theta}_{w_c}^\top \boldsymbol{\theta}_{w_t}) - \sum_{i=1}^{p} \log \left(1 - \sigma(\boldsymbol{\theta}_{w_i}^\top \boldsymbol{\theta}_{w_t})\right) \tag{1}$$

with $\boldsymbol{\theta}_{w_t}$ and $\boldsymbol{\theta}_{w_c}$ the embeddings of the target and context nodes (or words), $w_i$ the $i$-th negative sample drawn from a noise distribution, $p$ the number of negative samples, and $\sigma$ the sigmoid function.

**Knowledge-graph embedding**   RDF2vec [Ristoski and Paulheim, 2016] adapts SGNS to multi-relational graphs by simply adding the relations to the generated sentences.

More advanced methods model relations as geometric transformations in the embedding space. These *triple-based methods*, inspired by SGNS, represent the plausibility of a triple given the embeddings $\boldsymbol{\theta}_h, \boldsymbol{w}_r, \boldsymbol{\theta}_t$ of the entities and relation with a scoring function $f(h, r, t)$ often written as

Scoring function
$$f(h, r, t) = -\text{sim}(\phi(\boldsymbol{\theta}_h, \boldsymbol{w}_r), \boldsymbol{\theta}_t) \tag{2}$$

where $\phi$ is a model-specific relational operator, and sim a similarity function. The embeddings are optimized by gradient descent to maximize the score of positive triples, and minimize that of negative ones. A possible loss function is the binary cross-entropy loss [Ali et al., 2021a]

$$\mathcal{L}_{\text{BCE}} = -\log \sigma(f(h, r, t)) - \sum_{i=1}^{p} \log \left(1 - \sigma(f(h_i', r, t_i'))\right) \tag{3}$$

which boils down to SGNS for $f(h, r, t) = \boldsymbol{\theta}_h^\top \boldsymbol{\theta}_t$. These models strive to align, for positive triples, the tail embedding $\boldsymbol{\theta}_t$ with the "relationally" transformed head embedding $\phi(\boldsymbol{\theta}_h, \boldsymbol{w}_r)$. The challenge is to design a clever $\phi$ operator to model complex patterns in the data, like hierarchies, compositions, or symmetries. Indeed some relations are one-to-one (people only have one biological mother), well represented by a translation [Bordes et al., 2013], while others are many-to-one (for instance many person were BornIn Paris), calling for $\phi$ to contract distances [Wang et al., 2017]. Many models explore different parametrizations, among which MuRE [Balazevic et al., 2019a], RotatE [Sun et al., 2019], or QuatE [Zhang et al., 2019] have good performance [Ali et al., 2021a]. This framework also includes models like DistMult [Yang et al., 2015], ComplEx [Trouillon et al., 2016], or TuckER [Balazevic et al., 2019b], that implicitly perform tensor factorizations.

**Embedding propagation** To smooth computed embeddings, CompGCN [Vashishth et al., 2020] introduces the idea of propagating knowledge-graph embeddings using the relational operator $\phi$, but couples it with learnable weights and a non-linearity. REP [Wang et al., 2022] simplifies this framework by removing weight matrices and non-linearities. Rossi et al. [2022] also use Feature Propagation, but to impute missing node features in graphs. Albooyeh et al. [2020] incorporate propagation *within* the standard link prediction pipeline, with negative sampling and gradient descent on standard KGE loss functions.

## 2.2 Techniques for scaling graph algorithms

Various tricks help scale graph algorithms to the sizes we are interested in –millions of nodes.

**Graph partitioning** Scaling up graph computations, for graph embedding or more generally, often relies on breaking down graphs in subgraphs. Appendix E.1 presents corresponding prior work.

**Local subsampling** Other forms of data reduction can help to scale graph algorithms (*e.g.* based on message passing). Algorithms may subsample neighborhoods, as GraphSAGE [Hamilton et al., 2017] that selects a fixed number of neighbors for each node on each layer, or GraphSAINT [Zeng et al., 2020] that samples overlapping subgraphs through random walks, for supervised GNN training via node classification. Cluster-GCN [Chiang et al., 2019] restricts the neighborhood search within clusters, obtained by classic clustering algorithms, to improve computational efficiency. MariusGNN [Waleffe et al., 2023] uses an optimized data structure for neighbor sampling and GNN aggregation. TIGER [Wang et al., 2024] proposes a slice-and-cache procedure to optimize the subgraph extraction for large-scale inductive GNN training on knowledge graphs.

**Multi-level techniques** Multi-level approaches, such as HARP [Chen et al., 2018], GraphZoom [Deng et al., 2020] or MILE [Liang et al., 2021], coarsen the graph, compute embeddings on the smaller graph, and project them back to the original graph.

## 2.3 Scaling knowledge-graph embedding

**Parallel training** Many approaches scale triple-level stochastic solvers by distributing training across multiple workers, starting from the seminal PyTorch-BigGraph (PBG) [Lerer et al., 2019] that splits the triples into buckets based on the partitioning of the entities. The challenge is then to limit overheads and communication costs coming from 1) the additional data movement incurred by embeddings of entities occurring in several buckets 2) the synchronization of global trainable parameters such as the relation embeddings. For this, DGL-KE [Zheng et al., 2020] reduces data movement by using sparse relation embeddings and METIS graph partitioning [Karypis and Kumar, 1997] to distribute the triples across workers. HET-KG [Dong et al., 2022] further optimizes distributed training by preserving a copy of the most frequently used embeddings on each worker, to reduce communication costs. These "hot-embeddings" are periodically synchronized to minimize inconsistency. SMORE [Ren et al., 2022] leverages asynchronous scheduling to overlap CPU-based data sampling, with GPU-based embedding computations. Algorithmically, it contributes a rejection sampling strategy to generate the negatives at low cost. GraphVite [Zhu et al., 2019] accelerates SGNS for graph embedding by both parallelizing random walk sampling on multiple CPUs, and negative sampling on multiple GPUs. Marius [Mohoney et al., 2021] reduces synchronization overheads by opting for asynchronous training of entity embeddings with bounded staleness, and minimizes IO with partition caching and buffer-aware data ordering. GE2 [Zheng et al., 2024] improves data swap

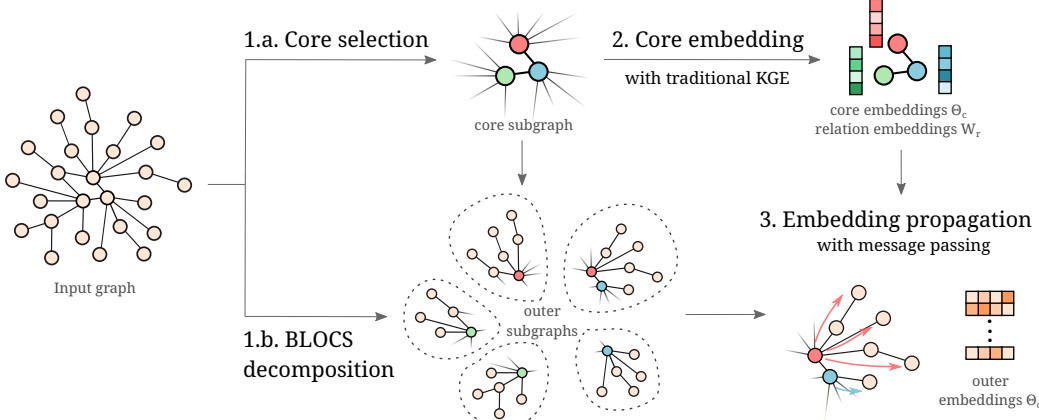

Figure 1: **SEPAL's embedding pipeline.** First, a core subgraph is extracted from the input knowledge graph (step *1.a*). BLOCS then subdivides this input knowledge graph into outer subgraphs (step *1.b*). Next, the core subgraph is embedded using traditional KGE models, which generate vector representations for both core entities and relations (step *2.*). Finally, these embeddings are propagated with message passing to each outer subgraph successively (step *3.*).

between CPU and multiple GPUs. Finally, the LibKGE [Broscheit et al., 2020] Python library also supports parallel training and includes GraSH [Kochsiek et al., 2022], an efficient hyperparameter optimization algorithm for large-scale KGE models.

**Parameter-efficient methods**   Other approaches reduce GPU memory pressure by limiting the number of parameters. NodePiece [Galkin et al., 2022] and EARL [Chen et al., 2023] embed a subset of entities and train an encoder to compute the embeddings of the other entities. However, their tokenization step is costly, and they have not been demonstrated on graphs larger than 2.5M nodes.

## 3   SEPAL: revisiting knowledge-graph embedding optimization

Most work on scaling knowledge-graph embedding has focused on efficient parallel computing to speed up stochastic optimization. We introduce a different approach, SEPAL, which changes how embeddings are computed, enforcing a more global structure, beneficial for downstream tasks, while avoiding much of the optimization cost. To that end, SEPAL proceeds in three steps (Figure 1):

1. separates the graph in a core and a set of connected overlapping outer subgraphs that cover the full graph;
2. uses a classic KGE model to optimize the embeddings of the core entities and relations;
3. propagates the embeddings from the core to the outer subgraphs, using a message-passing strategy preserving the relational geometry and ensuring embedding alignment within positive triples, with no further training.

SEPAL's key idea is to allocate more computation time to the more frequent entities and then use message passing to propagate embeddings at low cost to regions of the graph where they have not been computed yet. It departs from existing embedding propagation methods [Vashishth et al., 2020, Wang et al., 2022] that compute embeddings on the full graph and use propagation as post-processing to smooth them. SEPAL is compatible with any embedding model whose scoring function has the form given by Equation 2, examples of which are provided in Table 1.

### 3.1   Splitting large graphs into manageable subgraphs

Breaking up the graph into subgraphs is key to scaling up our approach memory-wise. Specifically, we seek a set of subgraphs that altogether cover the full graph but are individually small enough to fit on GPUs, to enable the subsequent GPU-based message passing.

**Core subgraph**   SEPAL first defines the *core* of a knowledge graph. The quality of the core embeddings is particularly important, as they serve as (fixed) boundary conditions during the propagation phase. Good relation embeddings are also key to structuring the propagation. To optimize this quality, two key factors must be considered during core selection: *1)* ensuring a dense core subgraph by selecting the most central entities and *2)* achieving full coverage of relation types. Yet, there can be a trade-off between these two objectives, hence, SEPAL offers two core selection procedures:

*Degree-based selection:* This simple approach selects the top entities by degree –with proportion $\eta_n \in (0,1)$– and keeps only the largest connected component of the induced subgraph. The resulting core is dense, which boosts performance for entity-centric tasks like feature enrichment (Appendix F.2.2). However, it does not necessarily contain all the relation types.

*Hybrid selection:* To ensure full relational coverage, this method combines two sampling strategies. First, it selects entities with the highest $\eta_n$ degrees. Second, for each relation type, it includes entities involved in edges with the highest $\eta_e$ degrees (where degree is the sum of the head and tail nodes' degrees). The union of these two sets forms the core, and if disconnected, SEPAL reconnects it by adding the necessary entities (details in Appendix F.2.2). Hyperparameters $\eta_n, \eta_e \in (0,1)$ are proportions of nodes and edges that control the core size.

Compared to degree-based selection, hybrid selection benefits tasks relying on relation embeddings, such as link prediction. However, its additional relation-specific edge sampling and reconnection steps can be computationally expensive for knowledge graphs with many relations. For disconnected input graphs, all connected components other than the largest one are added to the core subgraph.

**Outer subgraphs**   The next class of subgraphs that we generate –the *outer* subgraphs– aim at covering the rest of the graph. The purpose of these subgraphs demands the following requirements:

**R1: connected**   the subgraphs must be connected, to propagate the embeddings;
**R2: bounded size**   the subgraphs must have bounded sizes, to fit their embeddings in GPU memory;
**R3: coverage**   the union of the subgraphs must be the full graph, to embed every entity;
**R4: scalability**   extraction must run with available computing resources, in particular memory.

Extracting such subgraphs is challenging on large knowledge graphs. These are scale-free graphs with millions of nodes and no well-defined clusters [Leskovec et al., 2009]. They pose difficulties to partitioning algorithms. For instance, algorithms based on propagation, eigenvalues, or power iterations of the adjacency matrix [Raghavan et al., 2007, Shi and Malik, 2000, Newman, 2006] struggle with the presence of extremely high-degree nodes that make the adjacency matrix ill-conditioned. None of the existing partitioning algorithms satisfy our full set of constraints, and thus we devise our own algorithm, called BLOCS and described in detail in Appendix E. To satisfy the requirements despite these challenges, BLOCS creates *overlapping* subgraphs.

We contribute BLOCS, an algorithm designed to break large graphs into Balanced Local Overlapping Connected Subgraphs. The name summarizes the goals: 1) **Balanced**: BLOCS produces subgraphs of comparable sizes. $m$, the upper bound for subgraph sizes, is a hyperparameter. 2) **Local**: the subgraphs have small diameters. This locality property is important for the efficiency of SEPAL's propagation phase, as it reduces the number of propagation iterations needed to converge to the global embedding structure. 3) **Overlapping**: a given node can belong to several subgraphs. This serves our purpose because it facilitates information transfer between the different subgraphs during the propagation. 4) **Connected**: all generated subgraphs are connected.

BLOCS uses three base mechanisms to grow the subgraphs: *diffuse* (add all neighboring entities to the current subgraph), *merge* (merge two overlapping subgraphs), and *dilate* (add all unassigned neighboring entities to the current subgraph). There are two different regimes during the generation of subgraphs. First, few entities are assigned, and the computationally effective diffusion quickly covers a large part of the graph, especially entities that are close to high-degree nodes. However, once these close entities have been assigned, the effectiveness of diffusion drops because it struggles to reach entities farther away. For this reason, BLOCS switches from diffusion to dilation once the proportion of assigned entities reaches a certain threshold $h$ (a hyperparameter chosen $\approx .6$, depending on the dataset). By adding only unassigned neighbors to subgraphs, dilation drives subgraph growth towards unassigned distant entities. However, the presence of long chains can drastically slow down this regime because they make it add entities one by one. Some knowledge graphs have long chains, for

instance YAGO4.5 (see Diameter in Table 2). To tackle them, BLOCS switches back to diffusion for a few steps, with seeds taken inside the long chains.

BLOCS works faster on graphs that have small diameters, where most entities can be reached during the diffusion regime and fewer dilation steps are required (Appendix F.1).

## 3.2 Core optimization with traditional KGE models

Once the core subgraph is defined, SEPAL trains on GPU any compatible triple-based embedding model (DistMult, TransE, ...). This process generates embeddings for the core entities and relations, including inverse relations, added to ensure connectedness for the subsequent propagation step.

## 3.3 Outside the core: relation-aware propagation

Key to SEPAL's global consistency of embeddings and to computational efficiency is that it does not use contrastive learning and gradient descent for the outer entities. Instead, the final step involves an embedding propagation that is consistent with the KGE model (multiplication for DistMult, addition for TransE, ...) and preserves the relational geometry of the embedding space. To do so, SEPAL leverages the entity-relation composition function $\phi$ (given by Table 1) used by the KGE model, and the embeddings of the relations $\boldsymbol{w}_r$ trained on the core subgraph. From Equation 2 one can derive, for a given triple $(h, r, t)$, the closed-form expression of the tail embedding that maximizes the scoring function $\arg\max_{\boldsymbol{\theta}_t} f(h, r, t) = \phi(\boldsymbol{\theta}_h, \boldsymbol{w}_r)$. SEPAL uses this property to compute outer embeddings as consistent with the core as possible, by propagating from core entities with message passing.

First, the embeddings are initialized with $\quad \boldsymbol{\theta}_u^{(0)} = \begin{cases} \boldsymbol{\theta}_u, & \text{if entity } u \text{ belongs the core subgraph,} \\ \boldsymbol{0}, & \text{otherwise.} \end{cases}$

Then, each outer subgraph $\mathcal{S} \subset \mathcal{K}$ is loaded on GPU, merged with the core subgraph $\mathcal{C}$, and SEPAL performs $T$ steps of propagation ($T$ is a hyperparameter), with the following message-passing equations:

$$\text{Message:} \qquad \boldsymbol{m}_{v,u}^{(t+1)} = \sum_{(v,r,u)\in\mathcal{S}\cup\mathcal{C}} \phi(\boldsymbol{\theta}_v^{(t)}, \boldsymbol{w}_r) \tag{4}$$

$$\text{Aggregation:} \qquad \boldsymbol{a}_u^{(t+1)} = \sum_{v\in\mathcal{N}(u)} \boldsymbol{m}_{v,u}^{(t+1)} \tag{5}$$

$$\text{Update:} \qquad \boldsymbol{\theta}_u^{(t+1)} = \frac{\boldsymbol{\theta}_u^{(t)} + \alpha \boldsymbol{a}_u^{(t+1)}}{\left\| \boldsymbol{\theta}_u^{(t)} + \alpha \boldsymbol{a}_u^{(t+1)} \right\|_2} \tag{6}$$

where $\mathcal{N}(u)$ denotes the set of neighbors of outer entity $u$, $\mathcal{K}$ the set of positive triples of the graph, and $\alpha$ a hyperparameter similar to a learning rate. During updates, $\ell_2$ normalization projects embeddings on the unit sphere. With DistMult, this accelerates convergence by canceling the effect of neighbors that still have zero embeddings. Normalizing embeddings is a common practice in knowledge-graph embedding [Bordes et al., 2013, Yang et al., 2015], and SEPAL acts accordingly. During propagation, the core embeddings remain frozen.

## 4 Theoretical analysis: embedding alignment

**SEPAL minimizes a global energy via gradient descent** Proposition 4.1 shows that SEPAL with DistMult minimizes an energy that only accounts for the positive triples. The more aligned the embeddings within positive triples, the lower this energy. In self-supervised learning, negative sampling is needed to prevent embeddings from collapsing to a single point [Hafidi et al., 2022]. However, in our case, this oversmoothing is avoided thanks to the boundary conditions of fixed core-entities and relations embeddings, which act as "anchors" in the embedding space.

**Proposition 4.1** (Implicit Gradient Descent). *Let $\mathcal{E}$ be the "alignment energy" defined as*

$$\mathcal{E} = - \sum_{(h,r,t)\in\mathcal{K}} \langle \boldsymbol{\theta}_t, \phi(\boldsymbol{\theta}_h, \boldsymbol{w}_r) \rangle, \tag{7}$$

*with $\phi(\boldsymbol{\theta}_h, \boldsymbol{w}_r) = \boldsymbol{\theta}_h \odot \boldsymbol{w}_r$ being the DistMult relational operator. Then, SEPAL's propagation step amounts to a mini-batch projected gradient step descending $\mathcal{E}$ under the following conditions:*

1. *SEPAL uses DistMult as base KGE model;*
2. *the embeddings of relations and core entities remain fixed, and serve as boundary conditions;*
3. *the outer subgraphs act as mini-batches.*

*As a consequence, SEPAL converges towards a stationary point of $\mathcal{E}$ on the unit sphere.*

*Proof sketch.* For an outer entity $u$, the gradient update of its embedding $\boldsymbol{\theta}_u^{(t+1)} = \boldsymbol{\theta}_u^{(t)} - \eta \frac{\partial \mathcal{E}}{\partial \boldsymbol{\theta}_u^{(t)}}$ boils down to SEPAL's propagation equation, for a learning rate $\eta = \alpha$. See Appendix D.2 for detailed derivations. $\square$

**Analogy to eigenvalue problems** Appendix D.1 shows that SEPAL's propagation, when combined with DistMult, behaves like an Arnoldi iteration. This suggests that outer entity embeddings align with the dominant eigenvectors of an operator that captures both the graph structure and relational semantics (via fixed relation embeddings). In spectral graph theory, the eigenvectors of the Laplacian define the graph Fourier transform, and the leading ones correspond to the low frequencies. Similarly, in the SEPAL framework, the propagation computes 'low-frequency' representations for the outer entities, with the boundary conditions of fixed core embeddings.

**Queriability of embeddings** Assume that each entity $u \in \mathcal{V}$ is associated with a set of scalar features $\boldsymbol{x}_u \in \mathbb{R}^m$, such that $\boldsymbol{x}_{u,i}$ denotes the $i$-th property of $u$. We say that the embeddings support *queriability* with respect to property $i \in 1, \ldots, m$ if,

$$\exists f_i \in \mathcal{F} \quad \text{such that} \quad f_i(\boldsymbol{\theta}_u) \approx \boldsymbol{x}_{u,i}, \tag{8}$$

where $\mathcal{F}$ denotes a class of functions from $\mathbb{R}^d$ to $\mathbb{R}$ constrained by some regularity conditions. This queriability property is key to the embeddings' utility to downstream machine learning tasks.

In knowledge graphs, properties $\boldsymbol{x}_{u,i}$ that are explicitly represented by triples $(u, r_i, v_i)$ can be recovered via the scoring function $\phi$ used in the model, under the condition of well-aligned embeddings. Specifically, when such a relation $r_i$ exists, a natural querying function is $f_i : \boldsymbol{x} \mapsto \phi(\boldsymbol{x}, \boldsymbol{w}_{r_i})$, where $\boldsymbol{w}_{r_i}$ is the embedding of relation $r_i$ (we assume scalar embeddings for simplicity). This property holds only if the score of positive triples is maximized, implying that the corresponding facts have successfully been captured by the embeddings.

Proposition 4.1 shows that SEPAL minimizes an energy promoting alignment between embeddings within positive triples, which we argue leads to high queriability, since the scores of positive triples are maximized. In contrast, classic KGE methods use negative sampling to incorporate a supplementary constraint of local contrast between positive and negative triples. This adds discriminative power to the model, useful for link prediction, but can hinder queriability, as minimizing the score of negative triples may come at the expense of maximizing that of positive ones.

## 5 Experimental study: utility to downstream tasks

**Knowledge graph datasets** To compare large knowledge graphs of different sizes, we use Freebase [Bollacker et al., 2008], WikiKG90Mv2 [(an extract of Wikidata) Hu et al., 2020], and three generations of YAGO: YAGO3 [Mahdisoltani et al., 2014], YAGO4 [Pellissier Tanon et al., 2020], and YAGO4.5 [Suchanek et al., 2024]. We expand YAGO4 and YAGO4.5 into a larger version containing also the taxonomy, i.e., types and classes –which algorithms will treat as entities– and their relations. We discard numerical attributes and keep only the largest connected component (Appendix A.1). To perform an ablation study of SEPAL without BLOCS for which we need smaller datasets, we also introduce Mini YAGO3, a subset of YAGO3 built with the 5% most frequent entities. Knowledge graph sizes are reported in Figure 6.

**On real-world downstream regression tasks** We evaluate the embeddings as node features in downstream tasks [Grover and Leskovec, 2016, Cvetkov-Iliev et al., 2023, Robinson et al., 2024, Ruiz et al., 2024]. Specifically, we incorporate the embeddings in tables as extra features and measure the prediction improvement of a standard estimator in regression tasks. This setup allows us to compare the utility of knowledge graphs of varying sizes. Indeed, for a task, a suboptimal embedding of a larger knowledge graph may be more interesting than a high-quality embedding of a

smaller knowledge graph because the larger graph brings richer information, on more entities. We benchmark 4 downstream regression tasks [adapted from Cvetkov-Iliev et al., 2023]: Movie revenues, US accidents, US elections, and housing prices. Details are provided in Appendix A.2.

Larger knowledge graphs do bring value (Figure 2), as they cover more entities of the downstream tasks (Figure 4). For each source graph, SEPAL provides the best embeddings and is much more scalable (details in Figure 6). Considering performance/cost Pareto optimality across methods and source graphs (Figure 2a), SEPAL achieves the best performance with reduced cost, but the simple baseline FastRP also gives Pareto-optimal results, for smaller costs. Although FastRP discards the type of relation, it performs better than most dedicated KGE methods. Its iterations also solve a more global problem, like SEPAL (Appendix D.1).

We used DistMult as base model as it is a classic and good performer [Ruffinelli et al., 2020, Kadlec et al., 2017, Jain et al., 2020]. Figure 24 shows that SEPAL can speed up DistMult over 20 times for a given training configuration. For other scoring functions like RotatE and TransE, Figure 8 shows that SEPAL also improves the downstream performance of its base model.

**On WikiDBs tables** We also evaluate SEPAL on tables from WikiDBs [Vogel et al., 2024], a corpus of databases extracted from Wikidata. We build 42 downstream tables (26 classification tasks, and 16 regression tasks), described in Appendix A.2. Four of them are used as validation tasks, to tune hyperparameters, and the remaining 38 are used as test tables. Figure 3 presents the aggregated results on these 38 test tables, showing that, here also, applying SEPAL to very large graphs provides the best embeddings for downstream node regression and classification, and that SEPAL brings a decreased computational cost. Regarding the performance-cost tradeoff, FastRP is once again Pareto-optimal for small computation times, highlighting the benefits of global methods.

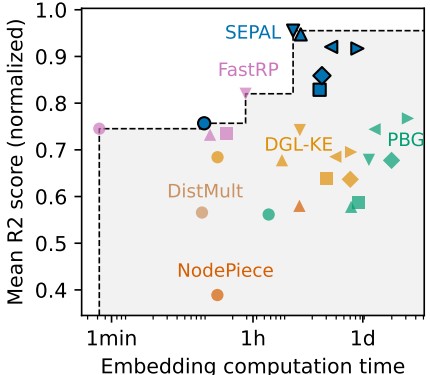

a) **Performance/cost Pareto frontier**

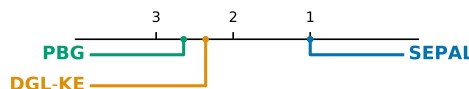

b) **Critical difference diagram (28 tasks)**

Figure 2: **Statistical performance on real-world tables. a) Pareto frontiers** of averaged normalized prediction scores with respect to embedding times (log-scale). **b) Critical difference diagrams** [Terpilowski, 2019] of average ranks among the three methods (SEPAL, PBG and DGL-KE) that scale to every knowledge-graph dataset. The ranks are averaged over all tasks; a task being defined as the combination of a downstream table and a source knowledge graph. SEPAL gets the best average downstream performance for each of the 7 source knowledge graphs. Figure 6 gives the detailed results for each table. Appendix B.1 details the metric used.

## 6 Discussion and conclusion

**Benefits of larger graphs** We have studied how to build general-knowledge feature enrichment from huge knowledge graphs. For this purpose, we have introduced a scalable method that captures more of the global structure than the classic KGE methods. Our results show the benefit of embedding larger graphs. There are two reasons to this benefit: (1) larger knowledge graphs can result in larger coverage of downstream entities (another important factor for this is the age of the dataset: recent ones have better coverage) (2) for two knowledge graphs with equal coverage, the larger one can result in richer representations because the covered entities have more context to learn from.

**Limitations** Our work focuses on embeddings for feature enrichment of downstream tables, an active research field [Cvetkov-Iliev et al., 2023, Ruiz et al., 2024, Robinson et al., 2024]. Another popular use case for embeddings is knowledge graph completion. However, this task is fundamentally different from feature enrichment: embeddings optimized for link prediction may not perform well

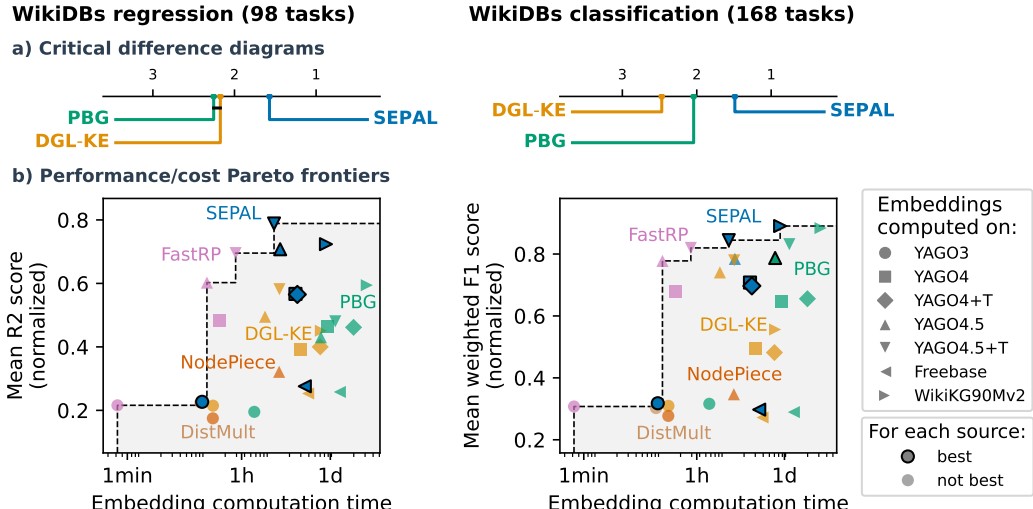

Figure 3: **Statistical performance on WikiDBs tables. a) Critical difference diagrams** of scalable methods. Black lines connect methods that are not significantly different. **b) Pareto frontiers** of averaged normalized prediction scores with respect to embedding times (log-scale). Figure 7 in Appendix C.2 provides the detailed results for each of the 38 test tables.

for feature enrichment, and vice versa [Ruffinelli and Gemulla, 2024]. Nevertheless, we also evaluate SEPAL on knowledge graph completion, for which we expect lower performances given that SEPAL does not locally optimize the contrast between positive and negative triples scores. Results reported in Appendix C.6 show that SEPAL sometimes performs lower than existing methods (DGL-KE, PBG) on this task, although no method is consistently better than the others for all datasets, and statistical tests show no significant differences (Figure 9).

**Conclusion: embeddings for downstream machine learning**    In this paper, we show how to optimize knowledge-graph embeddings for downstream machine learning. We propose a highly scalable method, SEPAL, and conduct a comprehensive evaluation on 7 knowledge graphs and 46 downstream tables showing that SEPAL both: (1) markedly improves predictive performance on downstream tasks and (2) brings computational-performance benefits –multiple-fold decreased train times and bounded memory usage– when embedding large-scale knowledge graphs. Our theoretical analysis suggests that SEPAL's strong performance on downstream tasks stems from its global optimization approach, resulting in better-aligned embeddings compared to classic methods based on negative sampling. SEPAL improves the quality of the generated node features when used for data enrichment in external (downstream) tasks, a setting that can strongly benefit from pre-training embeddings on knowledge bases as large as possible. It achieves this without requiring heavy engineering, such as distributed computing, and can easily be adapted to most KGE models.

Insights brought by our experiments go further than SEPAL. First, the method successfully exploits the asymmetry of information between "central" entities and more peripheral ones. Power-law distributions are indeed present on many types of objects, from words [Piantadosi, 2014] to geographical entities [Giesen and Südekum, 2011] and should probably be exploited for general-knowledge representations such as knowledge-graph embeddings. Second, and related, breaking up large knowledge graphs in communities is surprisingly difficult: some entities just belong in many (all?) communities, and others are really hard to reach. Our BLOCS algorithm can be useful for other graph engineering tasks, such as scaling message-passing algorithms or simply generating partitions. Finally, the embedding propagation in SEPAL appears powerful, and we conjecture it will benefit further approaches. First, it can be combined with much of the prior art to scale knowledge-based embedding. Second, it can naturally adapt to continual learning settings [Van de Ven et al., 2022, Hadsell et al., 2020, Biswas et al., 2023], which are important in knowledge-graph applications since knowledge graphs, such as Wikidata, are often continuously updated with new information (Appendix G.3).

## Acknowledgements

GV acknowledges support from ANR via grant TaFoMo (ANR-25-CE23-1822). This work is partly supported by Hi! PARIS and ANR/France 2030 program (ANR-23-IACL-0005).

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

# Appendix - Table of Contents

Table 2: Additional statistics on the knowledge graph datasets used. MSPL stands for Mean Shortest Path Length. The LCC column gives the percentage of entities of the graph that are in the largest connected component.

| | Maximum degree | Average degree | MSPL | Diameter | Density | LCC |
|---|---|---|---|---|---|---|
| Mini YAGO3 | 65 711 | 12.6 | 3.3 | 11 | 1e-4 | 99.98% |
| YAGO3 | 934 599 | 4.0 | 4.2 | 23 | 2e-6 | 97.6% |
| YAGO4.5 | 6 434 121 | 4.5 | 5.0 | 502 | 1e-7 | 99.7% |
| YAGO4.5+T | 6 434 122 | 5.0 | 4.0 | 5 | 1e-7 | 100% |
| YAGO4 | 8 606 980 | 12.9 | 4.5 | 28 | 3e-7 | 99.0% |
| YAGO4+T | 32 127 569 | 9.4 | 3.4 | 6 | 1e-7 | 100% |
| Freebase | 10 754 238 | 4.9 | 4.7 | 100 | 6e-8 | 99.1% |
| WikiKG90Mv2 | 37 254 176 | 12.8 | 3.6 | 98 | 1e-7 | 100.0% |

Table 3: Number of rows in the downstream tables.

| | US elections | Housing prices | US accidents | Movie revenues |
|---|---|---|---|---|
| Number of rows | 13 656 | 22 250 | 20 332 | 7 398 |

## A  Datasets

### A.1  Statistics on knowledge graph datasets

More statistics on the knowledge graph datasets are given in Table 2. Maximum and average degree figures highlight the scale-free nature of real-world knowledge graphs. The values for mean shortest path length (MSPL) and diameter (the diameter is the longest shortest path) are provided for the largest connected component (LCC). They are remarkably small, given the number of entities in the graphs. Contrary to other datasets, YAGO4.5, Freebase, and WikiKG90Mv2 contain 'long chains' of nodes, which account for their larger diameters.

The density $D$ is the ratio between the number of edges $|E|$ and the maximum possible number of edges:

$$D = \frac{|E|}{|\mathcal{V}|(|\mathcal{V}| - 1)}$$

where $|\mathcal{V}|$ denotes the number of nodes.

The LCC statistics show that for each knowledge graph, the largest connected component regroups almost all the entities.

### A.2  Downstream tables

**Real-world tables**  We use 4 real-world downstream tasks adapted from Cvetkov-Iliev et al. [2023] who also investigate knowledge-graph embeddings to facilitate machine learning. The specific target values predicted for each dataset are the following:

**US elections:**  predict the number of votes per party in US counties;

**Housing prices:**  predict the average housing price in US cities;

**US accidents:**  predict the number of accidents in US cities;

**Movie revenues:**  predict the box-office revenues of movies.

For each table, a log transformation is applied to the target values as a preprocessing step. Table 3 contains the sizes of these real-world downstream tables.

**WikiDBs tables**  WikiDBs contains 100,000 databases (collections of related tables), which altogether include 1,610,907 tables. We extracted 42 of those tables to evaluate embeddings. Here we describe the procedure used for table selection and processing.

Table 4: **Regression tables from WikiDBs.** The 'DB number' is the number of the database in WikiDBs from which the table was taken. Among these 16 regression tables, 2 are used for validation, and 14 for test.

| Table name | DB number | Value to predict | $N_{rows}$ | Set |
|---|---|---|---|---|
| Historical Figures | 62 826 | Birth date | 3 000 | Val |
| Geopolitical Regions | 66 610 | Land area | 2 324 | Val |
| Eclipsing Binary Star Instances | 3 977 | Apparent magnitude | 3 000 | Test |
| Research Article Citations | 14 012 | Publication date | 3 000 | Test |
| Drawings Catalog | 14 976 | Artwork height | 3 000 | Test |
| Municipal District Capitals | 19 664 | Population count | 2 846 | Test |
| Twinned Cities | 28 146 | Population | 1 194 | Test |
| Ukrainian Village Instances | 28 324 | Elevation (meters) | 3 000 | Test |
| Dissolved Municipality Records | 46 159 | Dissolution date | 3 000 | Test |
| Research Articles | 53 353 | Publication date | 3 000 | Test |
| Territorial Entities | 82 939 | Population count | 3 000 | Test |
| Artworks Inventory | 88 197 | Artwork width | 3 000 | Test |
| Business Entity Locations | 89 039 | Population count | 3 000 | Test |
| WWI Personnel Profiles | 89 439 | Birth date | 3 000 | Test |
| Registered Ships | 90 930 | Gross tonnage | 3 000 | Test |
| Poet Profiles | 94 062 | Death date | 3 000 | Test |

*Table selection:* Most of the WikiDBs tables are very small –typically a few dozen samples– so our first filtering criterion was the table size, which must be large enough to enable fitting an estimator. Therefore, we randomly sampled 100 tables from WikiDBs with sufficient sizes ($N_{rows} > 1,000$). Then we looked at each sampled table individually and kept those that could be used to define a relevant machine learning task (either regression or classification). We ended up with 16 regression tasks and 26 classification tasks.

*Table processing:* We removed rows with missing values, and reduced the size of large tables to keep evaluation tractable. For regression tables, we simply sampled 3,000 rows randomly (if the table had more than 3,000). We applied a log transformation to the target values to remove the skewness of their distributions. Before that, dates were converted into floats (by first converting them to fractional years, and then applying the transform $t \mapsto 2025 - t$). For classification tables, to reduce the dataset sizes while preserving both class diversity and balance, we downsampled the tables with the following procedure:

1. Class filtering: we set a threshold $r = \min(50, 0.9N_2)$, where $N_2$ is the cardinality of the second most populated class, and retained only the classes with more than $r$ occurrences.

2. Limit number of classes: if more than 30 classes remained after filtering, only the 30 most frequent were kept.

3. Downsampling: if the resulting table contained more than 3,000 rows, we sampled rows such that: (a) if $r \cdot N_{classes} \leq 3,000$, at least $r$ rows were sampled per class, (b) if $r \cdot N_{classes} > 3,000$, we sampled an approximately equal number of rows per class, fitting within the 3,000-row limit.

The specifications of the 42 tables extracted are given in Table 4 and Table 5.

## A.3 Entity coverage of downstream tables

**Entity coverage**  We define the coverage of table $T$ by knowledge graph $K$ as the proportion of downstream entities in $T$ that are described in $K$. Figure 4 gives the empirically measured coverage of the 46 downstream tables by the 8 different knowledge graphs used in our experimental study. It shows that larger and more recent knowledge graphs yield greater coverage.

**Entity matching**  Leveraging knowledge-graph embeddings to enrich a downstream tabular prediction task requires mapping the table entries to entities of the knowledge graph. We call this process *entity matching*.

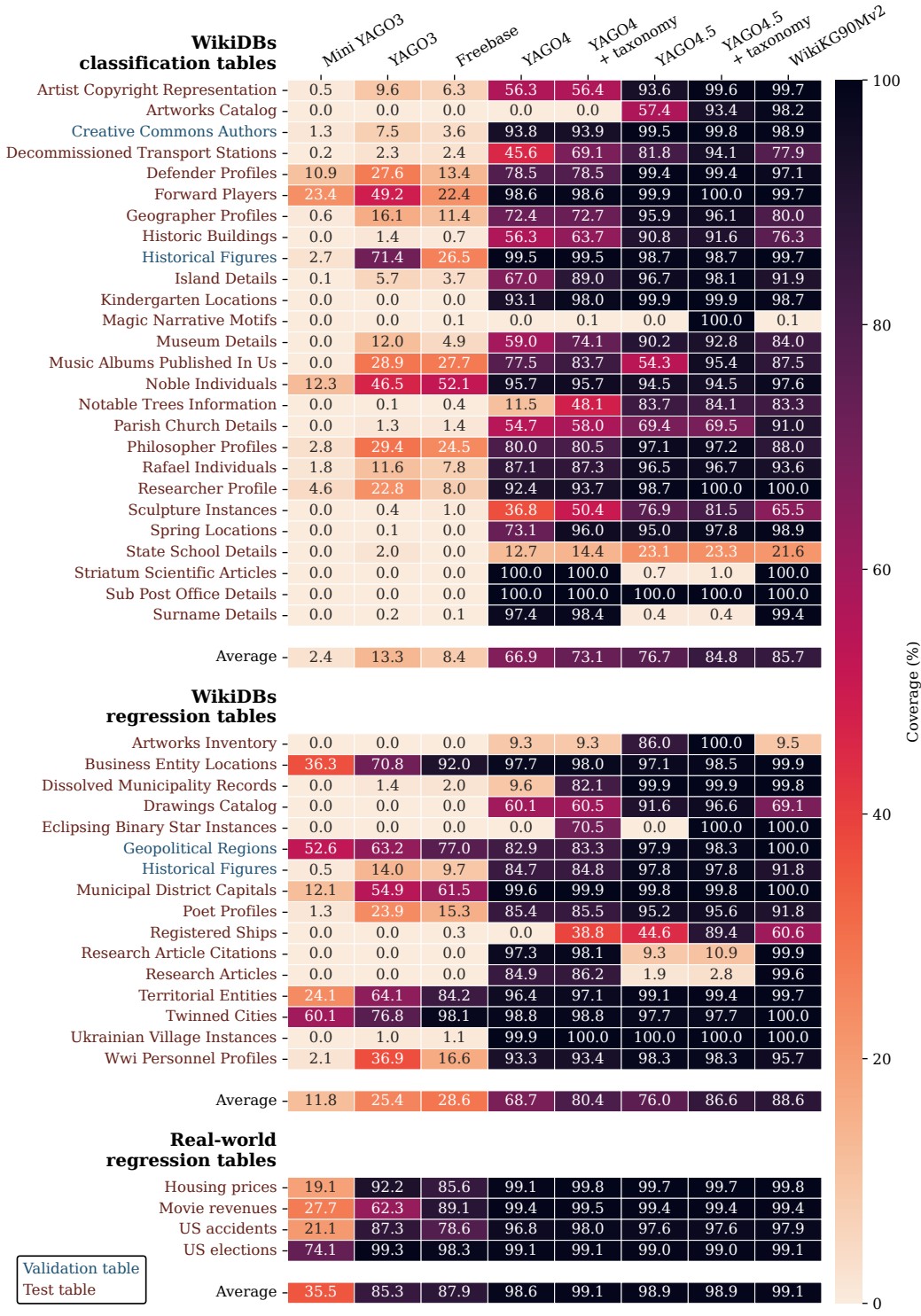

Figure 4: **Entity coverage of downstream tables.** Over the 46 downstream tables 4 are used for validation (in blue), and 42 are used for test (in maroon).

Table 5: **Classification tables from WikiDBs.** The 'DB number' is the number of the database in WikiDBs from which the table was taken. Among these 26 classification tables, 2 are used for validation, and 24 for test.

| Table name | DB number | Class to predict | $N_{rows}$ | $N_{classes}$ | Set |
|---|---|---|---|---|---|
| Creative Commons Authors | 9 510 | Gender | 2 999 | 2 | Val |
| Historical Figures | 73 376 | Profession | 1 044 | 5 | Val |
| Historic Buildings | 473 | Country | 2 985 | 30 | Test |
| Striatum Scientific Articles | 2 053 | Journal name | 2 986 | 30 | Test |
| Researcher Profile | 7 136 | Affiliated institution | 237 | 7 | Test |
| Decommissioned Transport Stations | 7 310 | Country | 2 983 | 30 | Test |
| Artist Copyright Representation | 7 900 | Artist occupation | 2 986 | 27 | Test |
| Forward Players | 15 542 | Team | 2 985 | 30 | Test |
| Rafael Individuals | 29 832 | Nationality | 2 966 | 12 | Test |
| Artworks Catalog | 30 417 | Artwork type | 2 991 | 17 | Test |
| Magic Narrative Motifs | 36 100 | Cultural origin | 2 993 | 12 | Test |
| Geographer Profiles | 42 562 | Language | 2 992 | 14 | Test |
| Surname Details | 47 746 | Language | 1 420 | 5 | Test |
| Sculpture Instances | 56 474 | Material used | 2 985 | 30 | Test |
| Spring Locations | 63 797 | Country | 2 981 | 30 | Test |
| Noble Individuals | 64 477 | Role | 2 987 | 30 | Test |
| Defender Profiles | 65 102 | Defender position | 2 998 | 5 | Test |
| Kindergarten Locations | 66 643 | Country | 2 998 | 4 | Test |
| Sub Post Office Details | 67 195 | Administrative territory | 2 986 | 30 | Test |
| State School Details | 70 780 | Country | 2 995 | 12 | Test |
| Notable Trees Information | 70 942 | Tree species | 2 992 | 19 | Test |
| Parish Church Details | 87 283 | Country | 2 993 | 15 | Test |
| Museum Details | 90 741 | Country | 2 986 | 30 | Test |
| Island Details | 92 415 | Country | 2 986 | 30 | Test |
| Philosopher Profiles | 97 229 | Language | 2 985 | 29 | Test |
| Music Albums Published in the US | 97 297 | Music Genre | 2 984 | 30 | Test |

For the 4 real-world tables, we performed the entity matching 'semi-automatically', following Cvetkov-Iliev et al. [2023]. The entries of these tables are well formatted and a small set of simple rules is sufficient to match the vast majority of entities. Human supervision was required for some cases of homonymy, for instance.

For the WikiDBs tables, we used the Wikidata Q identifiers (QIDs) included in the WikiDBs dataset to straightforwardly match the entities to WikiKG90Mv2[1], YAGO4, and YAGO4.5, which all provide the QIDs for every entity. YAGO4 also offers a mapping to the Freebase entities, which we used to match the Freebase entities to the WikiDBs tables. Additionally, both YAGO3 and YAGO4 provide mappings to DBpedia, which enabled us to obtain the matching for YAGO3.

## B  Evaluation methodology

### B.1  Downstream tasks

**Setting**   For each dataset, we use scikit-learn's Histogram-based Gradient Boosting Regression (*resp.* Classification) Tree [Pedregosa et al., 2011] as regression (*resp.* classification) estimator to predict the target value. The embeddings are the only features fed to the estimator, except for the US elections dataset for which we also include the political party. For embedding models outputting complex embeddings, such as RotatE, we simply concatenate real and imaginary parts before feeding them to the estimator. Figure 5 illustrates our evaluation setting.

The rows of the tables corresponding to entities not found in the knowledge graph are filled with NaNs as features for the estimator. This enables to compare the scores between different knowledge graphs (see Figure 2 and Figure 3) of different sizes to see the benefits obtained from embedding larger graphs, with better coverage of downstream entities (Figure 4).

---

[1]Entity mapping for WikiKG90Mv2 is provided at https://groups.google.com/g/open-graph-benchmark/c/R0SKtj9qQyE.

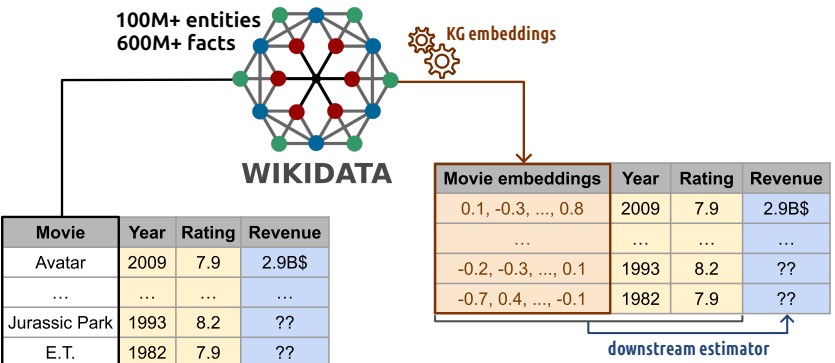

Figure 5: **Illustration of our evaluation setting for downstream tasks.** Embeddings computed on an external knowledge graph are used to enrich the features of a downstream table. The utility of these embeddings is measured by the improvement they bring to the predictions of a downstream estimator.

**Metrics**   The metric used for regression is the R2 score, defined as:

$$R^2 = 1 - \frac{\sum_{i=1}^{N}(y_i - \hat{y}_i)^2}{\sum_{i=1}^{N}(y_i - \bar{y})^2}$$

where $N$ is the number of samples (rows) in the target table, $y_i$ is the target value of sample $i$, $\hat{y}_i$ is the value predicted by the estimator, and $\bar{y}$ is the mean value of the target variable.

For classification, we use the weighted F1 score, defined as:

$$F1_{\text{weighted}} = \sum_{i=1}^{K} \frac{n_i}{N} \cdot F1_i$$

where $K$ is the number of classes, $n_i$ is the number of true instances for class $i$, $N = \sum_{i=1}^{K} n_i$ is the total number of samples, and $F1_i$ is the F1 score for class $i$.

To get the "*Mean score (normalized)*" reported on Figure 2 and Figure 3, we proceed as follows:

1. **Mean cross-validation score**: for each model[2] and evaluation table, the scores (R2 or weighted F1, depending on the task) are averaged over 5 repeats of 5-fold cross-validations.

2. **Normalized**: for each evaluation table, we divide the scores of the different models by the score of the best-performing model on this table. This makes the scores more comparable between the different evaluation tables.

3. **Average**: for each model, we average its scores across every evaluation table. The highest possible score for a model is 1. Getting a score of 1 means that the model beats every other model on every evaluation table.

**Validation/test split and hyperparameter tuning**   We use 4 of the 42 WikiDBs tables as validation data—2 for regression and 2 for classification tasks (see Figure 4). The remaining 38 WikiDBs tables, along with the 4 real-world tables, are used exclusively for testing.

The validation tables are used to tune hyperparameters and select the best-performing configuration for each method and each knowledge graph. Unless otherwise specified, all reported results are obtained on the test tables using these optimized configurations.

### B.2   Knowledge graph completion

We detail our experimental setup for the link prediction task:

---

[2]We define "*model*" as the combination of a method (*e.g.* DistMult, DGL-KE, etc.) and a knowledge graph on which it is trained.

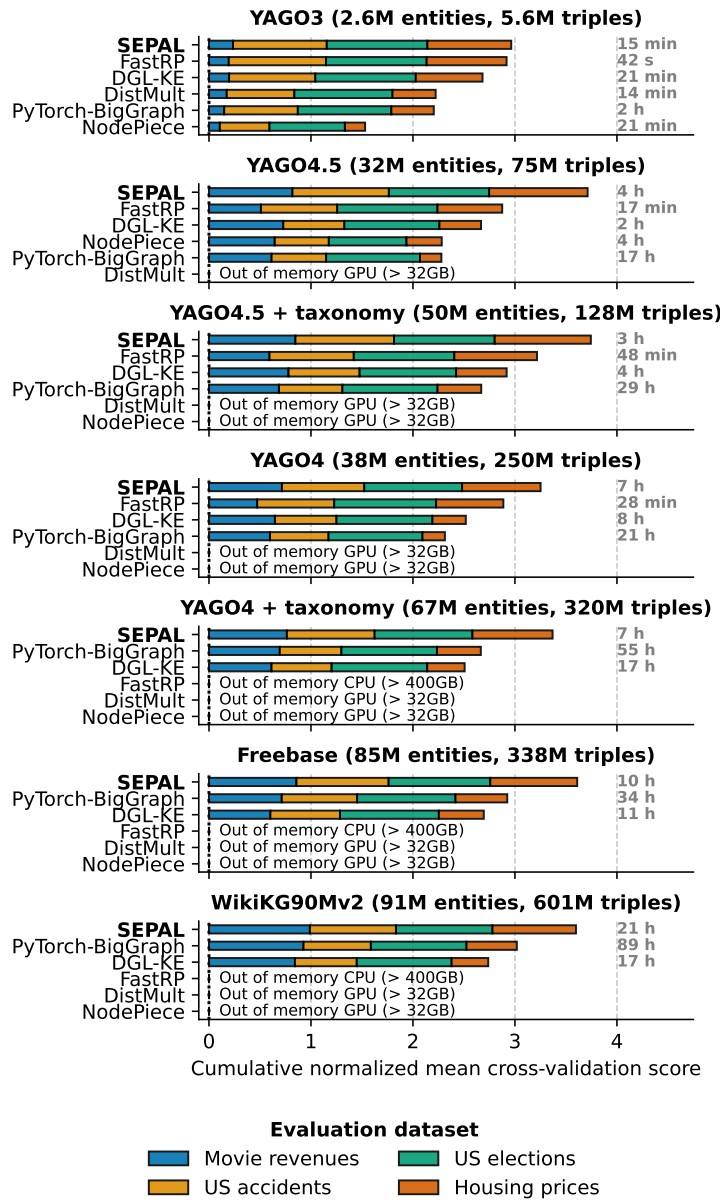

Figure 6: **Detailed results on real-world tables.** The "*Cumulative normalized mean cross-validation score*" reported is obtained by summing the normalized mean cross-validation scores. For an evaluation dataset, 1 corresponds to the best R2 score across all models; as there are 4 evaluation datasets, the highest possible score for a model is 4 (getting a score of 4 means that the model beats every model on every evaluation dataset). SEPAL, PyTorch-BigGraph, DGL-KE, and NodePiece use DistMult as base model. Embedding computation times are provided on the right-hand side of the figure. Figure 8 extends this figure with other embedding models.

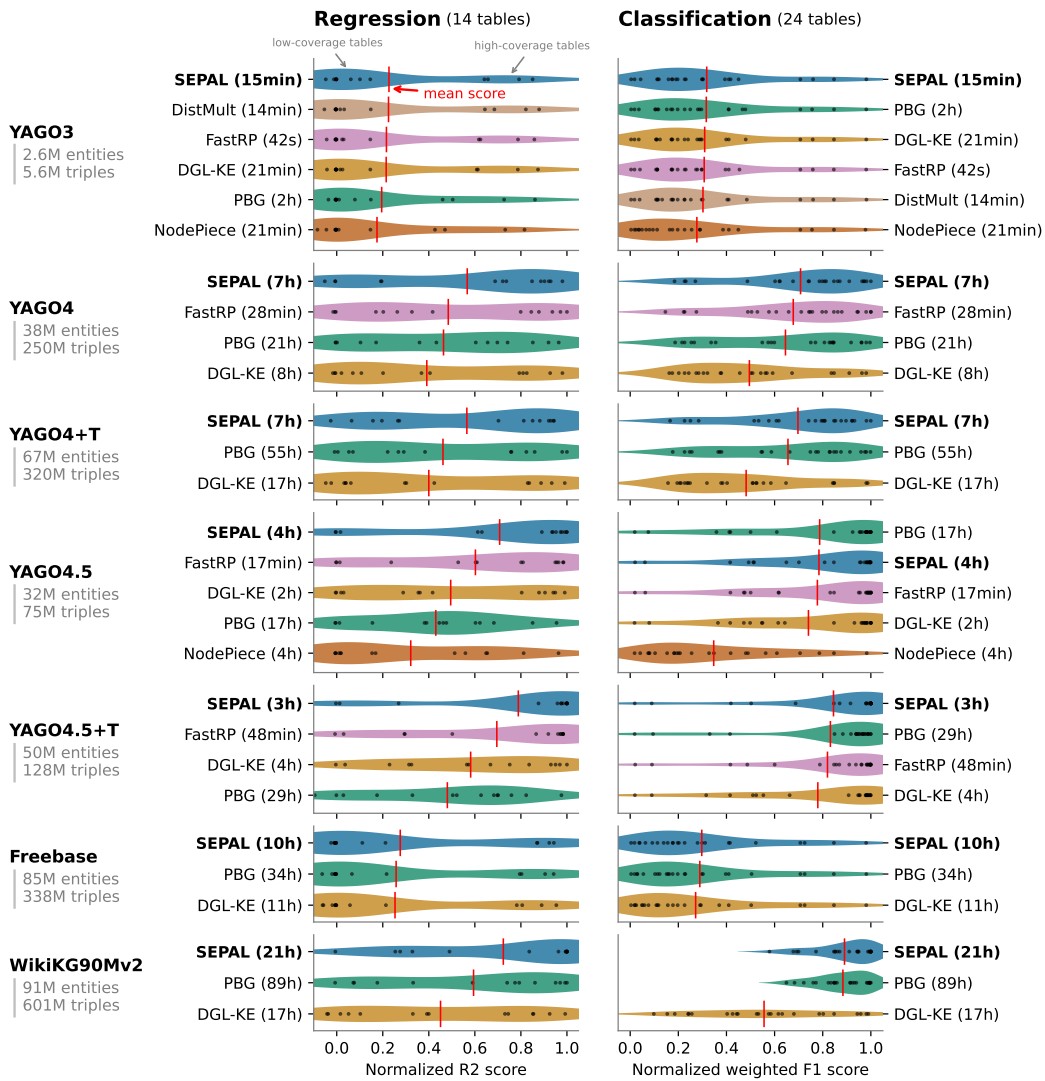

Figure 7: **Detailed results on WikiDBs tables.** Each black dot represents a downstream table. Red vertical lines indicate the mean score over all the tables. The methods appear in decreasing order of average score.

**Setting:** We evaluate models under the transductive setting: the missing links to be predicted connect entities already seen in the train graph. The task is to predict the tail entity of a triple, given its head and relation.

**Stratification:** We randomly split each dataset into training (90%), validation (5%), and test (5%) subsets of triples. During stratification, we ensure that the train graph remains connected by moving as few triples as required from the validation/test sets to the training set.

**Sampling:** Given the size of our datasets, sampling is required to keep link prediction tractable. For each evaluation triple, we sample 10,000 negative entities uniformly to produce 10,000 candidate negative triples by corrupting the positive.

**Filtering:** For tractability reasons, we report unfiltered results: we do not remove triples already existing in the dataset (which may score higher than the test triple) from the candidates.

**Ranking:** If several triples have the same score, we report realistic ranks (i.e., the expected ranking value over all permutations respecting the sort order; see PyKEEN documentation [Ali et al., 2021b]).

**Metrics:** We use three standard metrics for link prediction: the mean reciprocal rank (MRR), hits at $k$ (for $k \in \{1, 10, 50\}$) and mean rank (MR). Given the rankings $r_1, \ldots, r_n$ of the $n$ evaluation (validation or test) triples:

$$\text{MRR} = \frac{1}{n}\sum_{i=1}^{n}\frac{1}{r_i}, \qquad \text{Hits@}k = \frac{1}{n}\sum_{i=1}^{n}1_{r_i \leq k}, \qquad \text{MR} = \frac{1}{n}\sum_{i=1}^{n}r_i.$$

### B.3 Experimental setup

**Baseline implementations** In our empirical study, we compare SEPAL to DistMult, NodePiece, PBG, DGL-KE, and FastRP. We use the PyKEEN [Ali et al., 2021b] implementation for DistMult and NodePiece, and the implementations provided by the authors for the others. PBG was trained on 20 CPU nodes, and DGL-KE was allocated 20 CPU nodes and 3 GPUs; both methods were run on a single machine. The version of NodePiece we use for datasets larger than Mini YAGO3 is the ablated version, where nodes are tokenized only from their relational context (otherwise, the method does not scale on our hardware). For PBG, DGL-KE, NodePiece, and FastRP we used the hyperparameters provided by the authors for datasets of similar sizes. SEPAL and DistMult's hyperparameters were tuned on the validation sets presented in Appendix B.1 and Appendix B.2.

For all the baseline clustering algorithms, we used the implementations from the igraph package [Csardi, 2013] except for METIS, HDRF and Spectral Clustering. For METIS, we used the torch-sparse implementation, for Spectral Clustering, the scikit-learn [Pedregosa et al., 2011] implementation, and for HDRF, we used the C++ implementation from this repository.

**Computer resources** For PBG and FastRP, experiments were carried out on a machine with 48 cores and 504 GB of RAM. DistMult, DGL-KE, NodePiece, and SEPAL were trained on Nvidia V100 GPUs with 32 GB of memory, and 20 CPU nodes with 252 GB of RAM. The clustering benchmark was run on a machine with 88 CPU nodes and 504 GB of RAM.

## C Additional evaluation of SEPAL

### C.1 Table-level downstream results on real-world tables

Figure 6 shows the detailed prediction performance on the downstream tasks. SEPAL not only scales well to very large graphs (computing times markedly smaller than Pytorch-BigGraph), but also creates more valuable node features for downstream tasks.

### C.2 Table-level downstream results on WikiDBs tables

Figure 7 shows the detailed downstream tasks results for the WikiDBs test tables. For regression, SEPAL is the best performer on average for all of the 7 knowledge graphs. For classification, SEPAL is the best method on 6 of 7 knowledge graphs (narrowly beaten by PBG on YAGO4.5). Interestingly, FastRP, despite being very simple and not even accounting for relations, is a strong performer and beats more sophisticated methods like PBG and DGL-KE on many tasks. This is consistent with our *queriability* analysis in section 4 concluding that global methods, such as FastRP and SEPAL, are more suitable for downstream tasks, than local methods (such as PBG and DGL-KE).

For some knowledge graphs (especially Freebase and YAGO3), we can see two modes in the scores distribution, corresponding to the tables with low and high coverages (Figure 4).

### C.3 SEPAL combined with more embedding models

Figure 8 extends the results of Figure 6 by adding TransE and RotatE, alone and combined with SEPAL, as well as ablation studies results of SEPAL combined with METIS or ablated from BLOCS. These results show that SEPAL systematically improves upon its base model, whether it be DistMult RotatE or TransE. For a fair comparison, we ran RotatE with embedding dimension $d = 50$, as it outputs complex embeddings having twice as many parameters. For other models, we use $d = 100$.

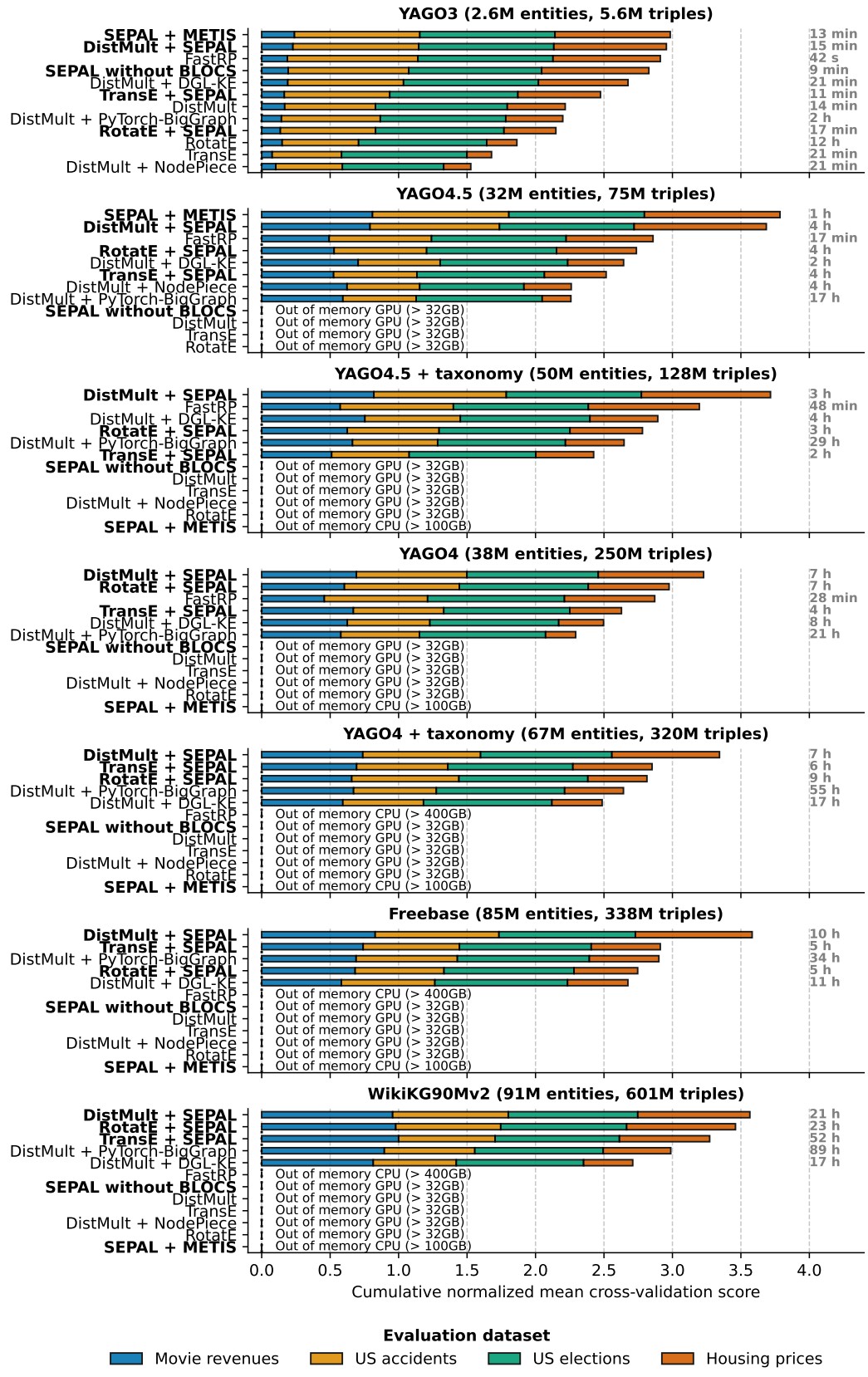

Figure 8: **Performance on real-world downstream tables.**

Table 6: Results for link prediction. Best in bold, second underlined.

| | YAGO3 | YAGO4.5 | YAGO4.5+T | YAGO4 | YAGO4+T | Freebase | WikiKG90Mv2 | Average |
|---|---|---|---|---|---|---|---|---|
| **a. MRR** | | | | | | | | |
| DistMult | **0.8049** | - | - | - | - | - | - | - |
| NodePiece | 0.2596 | 0.4456 | | - | - | - | - | - |
| PBG | 0.5581 | 0.5539 | 0.5688 | **0.6406** | **0.6224** | **0.7389** | **0.6325** | **0.6165** |
| DGL-KE | 0.7284 | **0.6200** | **0.6469** | 0.2372 | 0.2570 | 0.3017 | 0.3202 | 0.4445 |
| SEPAL | 0.6501 | 0.5537 | 0.5646 | 0.4726 | 0.477 | 0.5378 | 0.5291 | 0.5407 |
| **b. Hits@1** | | | | | | | | |
| DistMult | **0.7400** | - | - | - | - | - | - | - |
| NodePiece | 0.1735 | 0.3449 | | - | - | - | - | - |
| PBG | 0.5000 | 0.4939 | 0.4977 | **0.5494** | **0.5416** | **0.7015** | **0.5568** | **0.5487** |
| DGL-KE | 0.6663 | **0.5511** | **0.5733** | 0.1642 | 0.1892 | 0.2498 | 0.2416 | 0.3765 |
| SEPAL | 0.5412 | 0.4913 | 0.4922 | 0.3746 | 0.3755 | 0.4824 | 0.4502 | 0.4582 |
| **c. Hits@10** | | | | | | | | |
| DistMult | **0.9059** | - | - | - | - | - | - | - |
| NodePiece | 0.4388 | 0.6379 | | - | - | - | - | - |
| PBG | 0.6562 | 0.6642 | 0.7021 | **0.803** | **0.7662** | **0.8053** | **0.7693** | **0.7380** |
| DGL-KE | 0.8293 | **0.7446** | **0.7821** | 0.3786 | 0.3842 | 0.3964 | 0.4694 | 0.5692 |
| SEPAL | 0.8394 | 0.6650 | 0.6871 | 0.6573 | 0.6778 | 0.6398 | 0.6739 | 0.6915 |
| **d. Hits@50** | | | | | | | | |
| DistMult | **0.9504** | - | - | - | - | - | - | - |
| NodePiece | 0.7358 | 0.8112 | | - | - | - | - | - |
| PBG | 0.7259 | 0.7734 | 0.8136 | **0.8891** | **0.8514** | **0.8541** | **0.8531** | **0.8229** |
| DGL-KE | 0.8873 | **0.8171** | **0.8748** | 0.6037 | 0.5968 | 0.5547 | 0.654 | 0.7126 |
| SEPAL | 0.9204 | 0.7698 | 0.7786 | 0.7661 | 0.8068 | 0.7476 | 0.7805 | 0.7957 |
| **e. MR** | | | | | | | | |
| DistMult | **64.19** | - | - | - | - | - | - | - |
| NodePiece | 154.7 | **263.2** | - | - | - | - | - | - |
| PBG | 820.9 | 408.0 | 300.3 | **117.1** | **203.5** | 243.4 | 227.3 | 331.5 |
| DGL-KE | 187.7 | 624.1 | 219.9 | 224.9 | 271.5 | **227.0** | **186** | **277.3** |
| SEPAL | 95.87 | 270.4 | **206.0** | 553.0 | 363.2 | 357.3 | 365.5 | 315.9 |

## C.4 More baselines on the Freebase dataset

The Freebase dataset has been used in several previous works. To further validate SEPAL's performance, we compare it to additional baselines from the large-scale KGE literature: GraSH [Kochsiek et al., 2022], an efficient hyperparameter optimization framework for large-scale KGEs, and SMORE [Ren et al., 2022], a scalable KGE method supporting single-GPU training and multi-hop reasoning.

We train both of these baselines with DistMult as the base embedding method, on Freebase, using the authors' released configurations for this dataset. Table 7 reports the normalized scores on four real-world regression tasks, along with total training time.

We trained SMORE for 1 million iterations on one GPU, following the authors' configuration. Its relatively low performance here suggests that longer training could improve results, but under a 6-hour budget, SEPAL is substantially better.

GraSH optimizes hyperparameters for link prediction using successive halvings to discard unpromising configurations at low cost. Results show that, after 33 hours of hyperparameter search on one GPU, GraSH produces better embeddings than other baselines. However, SEPAL remains the best performer on all tasks, and also the fastest method.

Table 7: **Comparison to additional baselines on Freebase.** Normalized mean cross-validation scores on real-world downstream tasks, along with total training time. SEPAL outperforms all baselines while being the fastest to train.

| Method | Housing prices | Movie revenues | US accidents | US elections | Time |
|---|---|---|---|---|---|
| PBG | 0.513 | 0.723 | 0.742 | 0.957 | 33h 42m |
| DGL-KE | 0.445 | 0.610 | 0.686 | 0.962 | 10h 50m |
| SMORE | 0.160 | 0.332 | 0.410 | 0.926 | 6h 15m |
| GraSH | 0.601 | 0.862 | 0.810 | 0.961 | 32h 33m |
| SEPAL | **0.868** | **0.880** | **0.953** | **1.000** | **5h 58m** |

Table 8: **Evaluation on Ruffinelli et al.'s benchmark.** Classification and regression results on FB15k-237 (14k entities, 272k triples), YAGO3-10 (123k entities, 1M triples), and Wikidata5M (4.8M entities, 21M triples). Best results for each task are in bold.

| Dataset | Method | Classification weighted F1 ↑ | Regression RSE ↓ |
|---|---|---|---|
| FB15k-237 | ComplEx (MTT) | 0.858 | **0.394** |
| | RotatE (MTT) | **0.890** | 0.573 |
| | KE-GCN | 0.829 | 0.501 |
| | SEPAL | 0.853 | 0.492 |
| YAGO3-10 | DistMult (MTT) | 0.746 | 0.472 |
| | TransE (MTT) | 0.723 | 0.441 |
| | KE-GCN | 0.700 | 0.398 |
| | SEPAL | **0.762** | **0.386** |
| Wikidata5M | TransE (STD) | – | 0.596 |
| | SEPAL | – | **0.568** |

## C.5 Evaluation on prior benchmark

Ruffinelli and Gemulla [2024] propose a benchmark for evaluating KGE methods on downstream classification and regression. It includes several KGE baselines, with varying training procedures, and the graph neural network KE-GCN for entity classification. For each knowledge graph (FB15k-237, YAGO3-10, Wikidata5M), they evaluate embeddings on downstream tasks created from entity attributes of the knowledge graph.

We put the evaluation on this benchmark in appendix because our goal in this paper is to evaluate external, real-world tasks (Figure 2) that are independent of a specific knowledge graph, to compare the benefits of diverse knowledge graphs. In contrast, the Ruffinelli et al. tasks are created artificially and associated with specific knowledge graphs, that are orders of magnitude smaller than those of our main study.

However, evaluating SEPAL on these datasets complements our main experiments and enables direct comparison with prior work. Thus, we used 128-dimensional embeddings to match one of the dimensions in Ruffinelli et al.'s hyperparameter search space. We also used the authors' released evaluation script for comparable results. For each knowledge graph, Table 8 reports:

1. The best KGE method for classification from Ruffinelli and Gemulla [2024];
2. The best KGE method for regression from Ruffinelli and Gemulla [2024];
3. KE-GCN results from Ruffinelli and Gemulla [2024];
4. SEPAL results.

The results show that, on YAGO3-10 and Wikidata5M, SEPAL achieves the best performance for both regression and classification, consistent with our main results on much larger knowledge graphs. On FB15k-237, SEPAL has weaker results, but this may be due to the dataset size (FB15k-237 has 14k entities, 185 to 6,500 times smaller than the graphs considered in our main evaluation). For these tiny graphs, which fall outside SEPAL's intended scope, SEPAL may not be adapted because the core becomes too small to learn good representations for the relations and core entities.

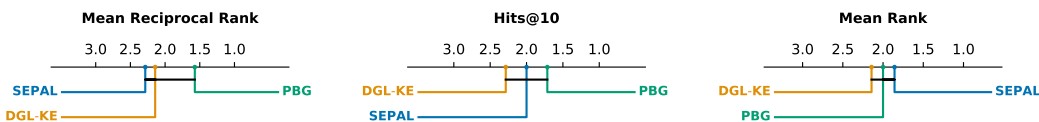

Figure 9: **Critical difference diagrams on link prediction metrics.** Statistical tests show no significant difference between methods at significance level $\alpha = 0.05$.

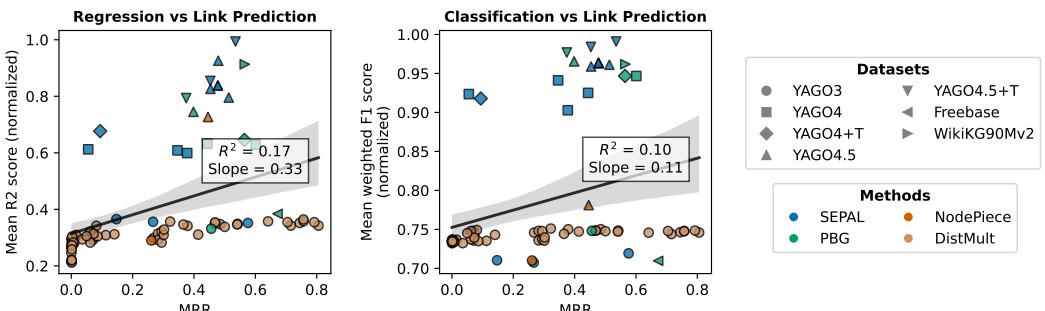

Figure 10: **Downstream task performance against link prediction performance**, on validation sets. Linear regression, with a 95% confidence interval. We plot the performance of models that share the same hyperparameters on the train graph (used for link prediction) and the full graph (used for downstream tasks).

## C.6 Evaluation on knowledge graph completion

### C.6.1 Link prediction results

Table 6 provides the link prediction results for the different metrics. It shows that, depending on the dataset, SEPAL is competitive with existing methods (DGL-KE, PBG) or not. However, none of the methods is consistently better than the others for all datasets. Following the analysis in section 4, these results are expected given that SEPAL does not enforce local contrast between positive and negative triples through contrastive learning with negative sampling, like other KGE methods do, but rather focuses on global consistency of the embeddings. Link prediction is, by essence, a local task, asking to discriminate efficiently between positives and negatives. For this purpose, it seems that the negative sampling, absent from SEPAL's propagation, plays a crucial role.

Nevertheless, Figure 9 shows that a Friedman test followed by Conover's post-hoc analysis [Conover and Iman, 1979, Conover, 1999] reveal no statistically significant differences (at significance level $\alpha = 0.05$) among the three methods that scale to all the knowledge graphs (SEPAL, PBG, and DGL-KE), as indicated by the critical difference diagrams where all methods are connected by a black line.

From a hyperparameter perspective, contrary to downstream tasks, the hybrid core selection strategy (with both node and relation sampling) yields better results than its simpler degree-based counterpart. This highlights different trade-offs between downstream tasks and link prediction. For link prediction, good relational coverage seems to count most, whereas for downstream tasks, the core density matters most (Figure 22).

### C.6.2 Downstream and link prediction performance weakly correlate

Figure 10 shows that embeddings performing well on downstream tasks do not necessarily perform well on link prediction, and vice versa. There is only a small positive correlation between these two performances: $R^2 \sim 0.1$.

## D Theoretical analysis

**Notations**  We use the following notations:

- $\boldsymbol{\Theta}^{(t)} \in \mathbb{R}^{n \times d}$ is the embedding matrix at step $t$, where each row is the embedding of an entity. Without loss of generality, we assume that the entities are ordered core first, then outer, so we can write $\boldsymbol{\Theta}^{(t)} = \begin{bmatrix} \boldsymbol{\Theta}_c \\ \boldsymbol{\Theta}_o^{(t)} \end{bmatrix}$ with $\boldsymbol{\Theta}_c \in \mathbb{R}^{n_c \times d}$ the embedding matrix of core (fixed) entities, and $\boldsymbol{\Theta}_o^{(t)} \in \mathbb{R}^{n_o \times d}$ the embedding matrix of outer (updated) entities at step $t$. $n_c$ and $n_o$ denote the number of core and outer entities, respectively, and $n_c + n_o = n$ is the total number of entities.

- $\boldsymbol{w}_r \in \mathbb{R}^d$: embedding of relation $r \in \mathcal{R}$ (fixed).

- $\boldsymbol{x}^{(t)} = \mathrm{vec}(\boldsymbol{\Theta}^{(t)}) = \left[ \boldsymbol{\Theta}_{1,1}^{(t)}, \ldots, \boldsymbol{\Theta}_{n,1}^{(t)}, \boldsymbol{\Theta}_{1,2}^{(t)}, \ldots, \boldsymbol{\Theta}_{n,2}^{(t)}, \ldots, \boldsymbol{\Theta}_{1,d}^{(t)}, \ldots, \boldsymbol{\Theta}_{n,d}^{(t)} \right]^\top \in \mathbb{R}^{nd}$: vectorization of the embedding matrix.

- $\boldsymbol{P} \in \mathbb{R}^{nd \times nd}$: global linear propagation matrix.

- $\boldsymbol{Q} \in \mathbb{R}^{nd \times nd}$: permutation matrix to reorder $\boldsymbol{x}$ into core and outer blocks.

- $\boldsymbol{y}^{(t)} = \boldsymbol{Q}\boldsymbol{x}^{(t)} = \left[ \boldsymbol{\theta}_1^\top; \ldots; \boldsymbol{\theta}_n^\top \right]^\top \in \mathbb{R}^{nd}$: reordered vector of embeddings. $\boldsymbol{y}^{(t)}$ can also be written as $\boldsymbol{y}^{(t)} = \begin{bmatrix} \boldsymbol{y}_c \\ \boldsymbol{y}_o^{(t)} \end{bmatrix}$.

- $\boldsymbol{M} = \boldsymbol{Q}(\boldsymbol{I} + \alpha\boldsymbol{P})\boldsymbol{Q}^{-1} \in \mathbb{R}^{nd \times nd}$: reordered update matrix. $\boldsymbol{M}$ can be written by block $\boldsymbol{M} = \begin{bmatrix} \boldsymbol{M}_{cc} & \boldsymbol{M}_{co} \\ \boldsymbol{M}_{oc} & \boldsymbol{M}_{oo} \end{bmatrix}$ where $\boldsymbol{M}_{oo}$ and $\boldsymbol{M}_{oc}$ are submatrices representing outer-to-outer and core-to-outer influences.

## D.1 Analysis of SEPAL's dynamic and analogies to eigenvalue problems

This section presents a theoretical analysis of the SEPAL propagation algorithm. We provide a series of intuitive and structural analogies to classic iterative methods in numerical linear algebra. Our goal is to contextualize the algorithm's behavior under various assumptions and shed light on its dynamic properties. The analysis is carried out in the case of DistMult, which simplifies the propagation rule due to its element-wise multiplication structure.

**SEPAL as power iteration (no normalization, no boundary conditions)**   We begin by analyzing the case without normalization or fixed embeddings. In this setting, with the vectorized embedding matrix $\boldsymbol{x}^{(t)} \in \mathbb{R}^{nd}$, the propagation equation (Equation 6) becomes:

$$\boldsymbol{x}^{(t+1)} = (\boldsymbol{I} + \alpha\boldsymbol{P})\boldsymbol{x}^{(t)}, \tag{9}$$

where $\boldsymbol{P} \in \mathbb{R}^{nd \times nd}$ encodes the linear update based on the knowledge graph structure and DistMult composition rule. The matrix $\boldsymbol{P}$ is block diagonal:

$$\boldsymbol{P} = \begin{bmatrix} \boldsymbol{P}^{(1)} & \boldsymbol{0} & \cdots & \boldsymbol{0} \\ \boldsymbol{0} & \boldsymbol{P}^{(2)} & \cdots & \boldsymbol{0} \\ \vdots & \vdots & \ddots & \vdots \\ \boldsymbol{0} & \boldsymbol{0} & \cdots & \boldsymbol{P}^{(d)} \end{bmatrix} \in \mathbb{R}^{nd \times nd}, \tag{10}$$

where each block $P^{(k)}$ corresponds to the $k$-th embedding dimension and has:

$$\boldsymbol{P}_{u,v}^{(k)} = \sum_{(v,r,u) \in \mathcal{K}} [\boldsymbol{w}_r]_k. \tag{11}$$

$P^{(k)}$ can be seen as a weighted adjacency matrix of the graph, whose weights are the $k$-th coefficients of the relation embeddings of the corresponding edges. Here, each $(v, r, u) \in \mathcal{K}$ contributes a rank-1 update to $\boldsymbol{P}$ based on $\boldsymbol{w}_r$.

Therefore, in this setting, the problem is separable with respect to the dimensions, so each dimension can be studied independently.

The recurrence in Equation 9 defines a classical *power iteration*. In general, it diverges unless the spectral radius $\rho(\boldsymbol{I} + \alpha\boldsymbol{P}) < 1$. In our setting, norms can grow arbitrarily because $\boldsymbol{P}$ contains non-normalized adjacency submatrices whose eigenvalues are only bounded by the maximum degree of the graph. In practice, the normalization (studied below) prevents the algorithm from diverging.

**With boundary conditions: non-homogeneous recurrence**  Now, we consider the SEPAL's setting where core entity embeddings are fixed and only outer embeddings evolve, still without normalization. We use the reordered vector of embeddings $\boldsymbol{y}^{(t)} = \boldsymbol{Q}\boldsymbol{x}^{(t)} \in \mathbb{R}^{nd}$, which can be written as follows:

$$\boldsymbol{y}^{(t)} = \begin{bmatrix} \boldsymbol{\theta}_1^{(t)} \\ \boldsymbol{\theta}_2^{(t)} \\ \vdots \\ \boldsymbol{\theta}_n^{(t)} \end{bmatrix} = \begin{bmatrix} \boldsymbol{y}_c \\ \boldsymbol{y}_o^{(t)} \end{bmatrix}, \tag{12}$$

where $\boldsymbol{\theta}_u^{(t)}$ is the embedding of entity $u$. The reordered update matrix $\boldsymbol{M} = Q(\boldsymbol{I} + \alpha\boldsymbol{P})Q^{-1}$ has the following block structure:

$$\boldsymbol{M} = \begin{bmatrix} \boldsymbol{M}_{11} & \cdots & \boldsymbol{M}_{1n} \\ \vdots & \ddots & \vdots \\ \boldsymbol{M}_{n1} & \cdots & \boldsymbol{M}_{nn} \end{bmatrix} \in \mathbb{R}^{nd \times nd}, \tag{13}$$

where each block $\boldsymbol{M}_{uv} \in \mathbb{R}^{d \times d}$ encodes how the embedding of entity $v$ contributes to the update of entity $u$. In the case of the DistMult scoring function, this block is diagonal and takes the form:

$$\boldsymbol{M}_{uv} = \begin{cases} \sum_{(v,r,u) \in \mathcal{K}} \alpha \cdot \text{diag}(\boldsymbol{w}_r) & \text{if } u \neq v, \\ \boldsymbol{I}_d + \sum_{(v,r,u) \in \mathcal{K}} \alpha \cdot \text{diag}(\boldsymbol{w}_r) & \text{otherwise,} \end{cases} \tag{14}$$

where $\text{diag}(\boldsymbol{w}_r)$ is the diagonal matrix with the relation embedding $\boldsymbol{w}_r \in \mathbb{R}^d$ on the diagonal.

Thus, $\boldsymbol{M}$ is a sparse block matrix, with each non-zero block being diagonal, and it linearly propagates information across entity embeddings via dimension-wise interactions determined by the DistMult model. Grouping core and outer entities together, we can also write $\boldsymbol{M} = \begin{bmatrix} \boldsymbol{M}_{cc} & \boldsymbol{M}_{co} \\ \boldsymbol{M}_{oc} & \boldsymbol{M}_{oo} \end{bmatrix}$.

This allows us to rewrite the propagation equation to account for the boundary conditions. We obtain:

$$\boldsymbol{y}_o^{(t+1)} = \boldsymbol{M}_{oo}\boldsymbol{y}_o^{(t)} + \boldsymbol{M}_{oc}\boldsymbol{y}_c, \tag{15}$$

where $\boldsymbol{M}_{oo}$ represents signal propagation between outer nodes, and $\boldsymbol{M}_{oc}$ encodes injection from the core nodes.

This is a *non-homogeneous linear recurrence*. If $\rho(\boldsymbol{M}_{oo}) \geq 1$, the outer embeddings diverge in norm. Nonetheless, the structure is analogous to forced linear systems such as:

$$\boldsymbol{y}_{t+1} = \boldsymbol{A}\boldsymbol{y}_t + \boldsymbol{b},$$

where the long-term behavior is driven by the balance between eigenvalues of $\boldsymbol{A}$ and direction of $\boldsymbol{b}$.

**Normalization and Arnoldi-type analogy**  SEPAL applies $\ell_2$ normalization after each update:

$$\boldsymbol{\theta}_u^{(t+1)} = \frac{\boldsymbol{\theta}_u^{(t)} + \alpha\boldsymbol{a}_u^{(t+1)}}{\|\boldsymbol{\theta}_u^{(t)} + \alpha\boldsymbol{a}_u^{(t+1)}\|_2}, \tag{16}$$

which constrains every embedding to the unit sphere. This *couples dimensions* and prevents simple linear analysis.

However, the recurrence

$$\boldsymbol{y}_o^{(t+1)} = \boldsymbol{M}_{oo}\boldsymbol{y}_o^{(t)} + \boldsymbol{M}_{oc}\boldsymbol{y}_c, \tag{17}$$

followed by blockwise normalization of $\boldsymbol{y}^{(t)}$ (with a $\ell_{2,\infty}$ mixed norm), shares structural similarities with the Arnoldi iteration [Arnoldi, 1951]:

- Successive embeddings span a Krylov subspace: after iteration $t$, without normalization, $\boldsymbol{y}_o^{(t)}$ belongs to $\mathcal{K}_t(\boldsymbol{A}, \boldsymbol{b}) = \text{span}\{\boldsymbol{b}, \boldsymbol{A}\boldsymbol{b}, \boldsymbol{A}^2\boldsymbol{b}, \dots, \boldsymbol{A}^{t-1}\boldsymbol{b}\}$, with $\boldsymbol{A} = \boldsymbol{M}_{oo}$ and $\boldsymbol{b} = \boldsymbol{M}_{oc}\boldsymbol{y}_c$, given that $\boldsymbol{y}_o^{(0)} = \boldsymbol{0}$.

- Core embeddings define the forcing direction ($\boldsymbol{b} = \boldsymbol{M}_{oc}\boldsymbol{y}_c$).
- Normalization serves as a regularizer, preventing divergence in norm.

The Arnoldi iteration is used to compute numerical approximations of the eigenvectors of general matrices, for instance, in ARPACK [Lehoucq et al., 1998]. This analogy suggests that the direction of outer embeddings stabilizes over time and aligns with a form of dominant generalized eigenvector of the propagation operator $\boldsymbol{M}$, conditioned on the core. Therefore, the embeddings produced by SEPAL's propagation encapsulate *global* structural information on the knowledge graph.

### D.2 Formal proof of Proposition 4.1

Gradient descent on $\mathcal{E}$ updates the embedding parameters $\boldsymbol{\theta}_u$ of an entity $u$ with

$$\boldsymbol{\theta}_u^{(t+1)} = \boldsymbol{\theta}_u^{(t)} - \eta \frac{\partial \mathcal{E}}{\partial \boldsymbol{\theta}_u^{(t)}} \tag{18}$$

where $\eta$ is the learning rate.

The mini-batch gradient satisfies

$$\frac{\partial \mathcal{E}}{\partial \boldsymbol{\theta}_u^{(t)}} = - \sum_{(h,r,t)\in\mathcal{B}} \frac{\partial \left\langle \boldsymbol{\theta}_t^{(t)}, \phi(\boldsymbol{\theta}_h^{(t)}, \boldsymbol{w}_r) \right\rangle}{\partial \boldsymbol{\theta}_u^{(t)}}$$

$$= - \sum_{\substack{(h,r,t)\in\mathcal{B} \\ h=u}} \frac{\partial \left\langle \boldsymbol{\theta}_t^{(t)}, \phi(\boldsymbol{\theta}_u^{(t)}, \boldsymbol{w}_r) \right\rangle}{\partial \boldsymbol{\theta}_u^{(t)}} - \sum_{\substack{(h,r,t)\in\mathcal{B} \\ t=u}} \frac{\partial \left\langle \boldsymbol{\theta}_u^{(t)}, \phi(\boldsymbol{\theta}_h^{(t)}, \boldsymbol{w}_r) \right\rangle}{\partial \boldsymbol{\theta}_u^{(t)}},$$

where $\mathcal{B}$ is the mini-batch. Note that the previous identity is still true if the knowledge graph contains self-loops, due to DistMult's scoring function being a product. Plugging the DistMult relational operator function $\phi(\boldsymbol{\theta}_h, \boldsymbol{w}_r) = \boldsymbol{\theta}_h \odot \boldsymbol{w}_r$ in the previous equation gives

$$\frac{\partial \mathcal{E}}{\partial \boldsymbol{\theta}_u^{(t)}} = - \sum_{\substack{(h,r,t)\in\mathcal{B} \\ h=u}} \frac{\partial \boldsymbol{\theta}_u^{(t)\top}(\boldsymbol{w}_r \odot \boldsymbol{\theta}_t^{(t)})}{\partial \boldsymbol{\theta}_u^{(t)}} - \sum_{\substack{(h,r,t)\in\mathcal{B} \\ t=u}} \frac{\partial \boldsymbol{\theta}_h^{(t)\top}(\boldsymbol{w}_r \odot \boldsymbol{\theta}_u^{(t)})}{\partial \boldsymbol{\theta}_u^{(t)}}$$

$$= - \sum_{\substack{(h,r,t)\in\mathcal{B} \\ h=u}} \boldsymbol{w}_r \odot \boldsymbol{\theta}_t^{(t)} - \sum_{\substack{(h,r,t)\in\mathcal{B} \\ t=u}} \boldsymbol{\theta}_h^{(t)} \odot \boldsymbol{w}_r$$

$$= - \sum_{\substack{(h,r,t)\in\mathcal{B} \\ h=u}} \boldsymbol{w}_r \odot \boldsymbol{\theta}_t^{(t)} - \sum_{\substack{(h,r,t)\in\mathcal{B} \\ t=u}} \phi(\boldsymbol{\theta}_h^{(t)}, \boldsymbol{w}_r).$$

Therefore, going back to Equation 18, we get that

$$\boldsymbol{\theta}_u^{(t+1)} = \boldsymbol{\theta}_u^{(t)} + \eta \underbrace{\sum_{\substack{(h,r,t)\in\mathcal{B} \\ t=u}} \phi(\boldsymbol{\theta}_h^{(t)}, \boldsymbol{w}_r)}_{\text{embedding propagation update for } \eta=\alpha \text{ and } \mathcal{B}=\mathcal{S}\cup\mathcal{C}} + \eta \sum_{\substack{(h,r,t)\in\mathcal{B} \\ h=u}} \boldsymbol{w}_r \odot \boldsymbol{\theta}_t^{(t)}. \tag{19}$$

We can see that the embedding propagation update only differs by a term that corresponds to the message passing from the tails to the heads. We did not include this term in our message-passing framework because we wanted SEPAL to adapt to any model whose scoring function has the form given by Equation 2. In practice, the propagation direction *tail* to *head* is already handled by the addition of inverse relations.

Therefore, a parallel can be drawn between: 1) the outer subgraphs and mini-batches, 2) the number of propagation steps $T$ and the number of epochs, 3) the hyperparameter $\alpha$ and the learning rate.

After each gradient step, we normalize the entity embeddings to enforce the unit norm constraint. This procedure corresponds to *projected gradient descent* on the sphere [Bertsekas, 1997], where each update is followed by a projection (via $\ell^2$ normalization) back onto the feasible set. The energy

function $\mathcal{E}$ (Equation 7) is composed of inner products and element-wise multiplications of smooth functions, thus it is smooth, and its gradient is Lipschitz continuous on the unit sphere. Under these conditions, the algorithm is guaranteed to converge to a stationary point of the constrained optimization problem [Bertsekas, 1997, Proposition 2.3.2]. The limit points of the optimization thus satisfy the first-order optimality conditions on the sphere.

# E    Presentation and analysis of BLOCS

## E.1    Prior work on graph partitioning

Scaling up computation on graph, for graph embedding or more generally, often relies on breaking down graphs in subgraphs. METIS [Karypis and Kumar, 1997], a greedy node-merging algorithm, is a popular solution. A variety of algorithms have also been developed to detect "communities", groups of nodes more connected together, often with applications on social networks: *Spectral Clustering* (SC) [Shi and Malik, 2000], the *Leading Eigenvector* (LE) method [Newman, 2006], the *Label Propagation Algorithm* (LPA) [Raghavan et al., 2007], the Louvain method [Blondel et al., 2008], the *Infomap* method [Rosvall and Bergstrom, 2008], and the *Leiden* method [Traag et al., 2019] which

---

**Algorithm 1** BLOCS

**Input:** Graph $\mathcal{G} = (V, E)$ with nodes $V$ and edges $E$, hyperparameters $h$ and $m$
**Output:** List of overlapping connected subgraphs
$\mathbb{S} \leftarrow \emptyset$  ▷ *list of subgraphs*
$U \leftarrow V$  ▷ *set of unassigned nodes*
**Step 1: Create subgraphs from super-spreaders' neighbors**
**for** each node $v \in V$ **do**
    **if** $deg(v) > 0.2\,m$ **then**
        $\mathbb{S}, U \leftarrow$ SplitNeighbors$(v, \text{max\_size} = 0.2\,m)$
    **end if**
**end for**
**Step 2: Assign nodes to subgraphs by diffusion**
**while** $|U| > (1-h)|V|$ **do**
    $k \leftarrow 0$  ;  $\mathcal{S}_0 \leftarrow \{\arg\max_{v \in U} deg(v)\}$  ▷ *start with unassigned node $v$ with highest degree*
    **while** $|\mathcal{S}_k| < 0.8\,m$ **do**
        $\mathcal{S}_{k+1} \leftarrow$ Diffuse$(\mathcal{S}_k)$  ;  $k \leftarrow k+1$
    **end while**
    Append $\mathcal{S}_{k-1}$ to $\mathbb{S}$, and update $U$  ▷ *$\mathcal{S}_{k-1}$ is the last subgraph smaller than $0.8\,m$*
**end while**
**Step 3: Merge small overlapping subgraphs**
$\mathbb{S}, U \leftarrow$ MergeSmallSubgraphs$(\mathbb{S}, \text{min\_size} = m/2)$
**Step 4: Dilation and diffusion until all entities are assigned**
$p \leftarrow 0$  ▷ *create new subgraphs by diffusion every 5 rounds, to tackle long chains*
**while** $|U| > 0$ **do**
    **if** $5$ divides $p$ and $p > 0$ **then**
        $i \leftarrow 0$
        **repeat**
            $k \leftarrow 0$  ;  $\mathcal{S}_0 \leftarrow \{\arg\max_{v \in U} deg(v)\}$
            **while** $|\mathcal{S}_k| < 0.8\,m$ **do**
                $\mathcal{S}_{k+1} \leftarrow$ Diffuse$(\mathcal{S}_k)$  ;  $k \leftarrow k+1$
            **end while**
            Append $\mathcal{S}_{k-1}$ to $\mathbb{S}$, and update $U$  ;  $i \leftarrow i+1$
        **until** $i = 10$
    **end if**
    $\mathbb{S} \leftarrow$ Dilate$(\mathbb{S})$  ;  $p \leftarrow p+1$
**end while**
**Step 5: Merge small overlapping subgraphs again**
$\mathbb{S}, U \leftarrow$ SystematicMerge$(\mathbb{S}, \text{min\_size} = 0.4\,m)$
**Step 6: Split subgraphs larger than $m$**
$\mathbb{S}, U \leftarrow$ SplitLargeSubgraphs$(\mathbb{S}, \text{max\_size} = m)$
$\mathbb{S}, U \leftarrow$ MergeSmallSubgraphs$(\mathbb{S}, \text{min\_size} = m/2)$
**Return:** $\mathbb{S}$, the set of overlapping subgraphs covering $\mathcal{G}$

---

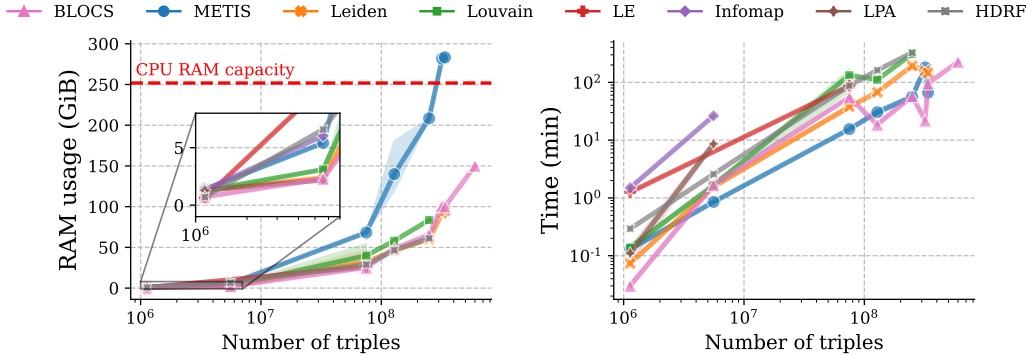

Figure 11: **Scalability of partitioning methods.** Memory usage and time for 8 knowledge graphs of various sizes. The "CPU RAM capacity" dashed line represents the CPU RAM of the machine we used to run SEPAL (we used a different machine with more RAM to run this benchmark, see Appendix B.3). Partitioning methods going beyond this limit thus cannot be combined with SEPAL on our hardware. BLOCS is the only method to scale up to WikiKG90Mv2, a knowledge graph with 601M triples. Leiden and METIS both caused memory errors on WikiKG90Mv2, while HDRF was too long for graphs larger than YAGO4 (our time limit was set to 333 minutes).

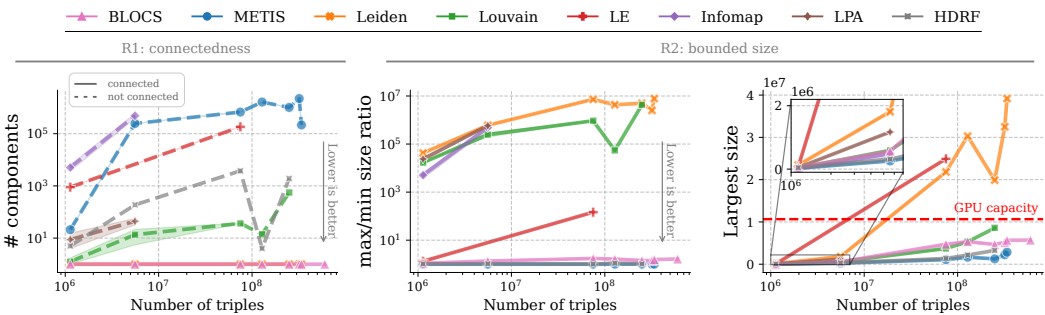

Figure 12: **Quality of partitioning methods.** Maximum number of connected components in one partition, ratio between the largest and smallest partition sizes, and size (number of entities) of the largest partition produced for knowledge graphs of various sizes. The "GPU capacity" dashed line represents the typical number of entities that can be loaded onto the GPU before causing a memory error. Methods producing partitions larger than this cannot be combined with SEPAL, since the partition's embeddings must fit in GPU memory. BLOCS and Leiden are the only methods to consistently return connected partitions (requirement R1). BLOCS, METIS, and HDRF are the only methods to control the partition size (requirement R2).

guarantees connected communities. LDG [Stanton and Kliot, 2012] and FENNEL [Tsourakakis et al., 2014] are streaming algorithms for very large graphs. Some algorithms have also been specifically tailored for power-law graphs: DBH [Xie et al., 2014] leverages the skewed degree distributions to reduce the communication costs, HDRF [Petroni et al., 2015] is a streaming partitioning method that replicates high-degree nodes first, and Ginger [Chen et al., 2019b] is a hybrid-cut algorithm that uses heuristics for more efficient partitioning on skewed graphs.

### E.2 Detailed algorithm and pseudocode

Algorithm 1 describes BLOCS pseudocode, and Figure 13 illustrates its base mechanisms. Below, we provide more details on subparts of the algorithm.

The function `SplitNeighbors` is designed to manage high-degree nodes by distributing their neighboring entities into smaller subgraphs. For each node whose degree exceeds a certain threshold, it groups its neighbors into multiple subgraphs smaller than this threshold. These new subgraphs also include the original high-degree node to maintain connectedness. By assigning the neighbors

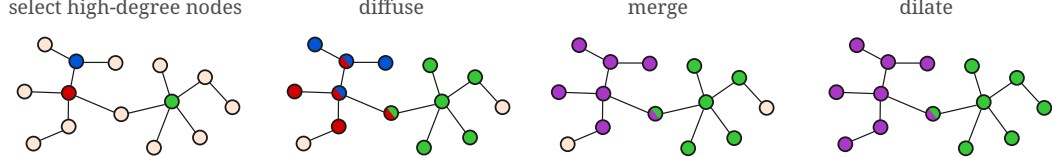

Figure 13: **Base mechanisms of the BLOCS algorithm.** BLOCS grows subgraphs from high-degree unassigned nodes using three main operations: *diffuse* adds all neighbors to the current subgraph; *merge* combines two overlapping subgraphs; and *dilate* adds only unassigned neighbors.

of the very high-degree nodes first, BLOCS ensures that subgraphs grow more progressively in the subsequent diffusion step.

The function `MergeSmallSubgraphs` takes a list of subgraphs and a minimum size threshold as input. It identifies subgraphs that are smaller than this given minimum size and merges them into larger subgraphs if they share nodes and if the size of their union remains below the maximum size $m$.

The function `SystematicMerge` is very similar to `MergeSmallSubgraphs`, but it does not check that the resulting subgraphs are smaller than $m$. Its objective is to eliminate all the subgraphs whose size is smaller than a given threshold (set to $0.4\,m$ in Algorithm 1). The subgraphs produced that are larger than $m$ are then handled by the function `SplitLargeSubgraphs`.

The function `SplitLargeSubgraphs` processes the list of subgraphs to break down overly large subgraphs while preserving connectivity. The function iterates over the subgraphs having more than $m$ nodes, subtracts the core, and computes the connected components. Then, it creates new subgraphs by grouping these connected components until the size limit $m$ is reached. As a result, outer subgraphs can be disconnected at this stage, but merging them with the core ensures their connectedness during embedding propagation.

### E.3 Benchmarking BLOCS against partitioning methods

First, we compare BLOCS to other graph partitioning, clustering, and community detection methods. Figure 11 and Figure 12 report empirical evaluation on eight knowledge graphs. BLOCS, METIS, and Leiden are the only approaches that scale to the largest knowledge graphs. Others fail due to excessive runtimes –our limit was set to $2 \cdot 10^4$ seconds. Compared to METIS, BLOCS is more efficient in terms of RAM usage while having similar computation times (Figure 11). Experimental results also show that classic partitioning methods fail to meet the connectedness and size requirements. Indeed, knowledge graphs are prone to yield disconnected partitions due to their scale-free nature: they contain very high-degree nodes. Such a node is hard to allocate to a single subgraph, and subgraphs without it often explode into multiple connected components. Our choice of overlapping subgraphs avoids this problem.

**Classic methods do not meet the requirements of SEPAL**     Here, we provide qualitative observations on the partitions produced by the different methods. We explain why they fail to meet our specific requirements.

**METIS**  is based on a multilevel recursive bisection approach, which coarsens the graph, partitions it, and then refines the partitions. It produces partitions with the same sizes; however, due to the graph structure, they often explode into multiple connected components, which is detrimental to the embedding propagation (see Appendix E.3).

**Louvain**  is based on modularity optimization. It outputs highly imbalanced communities, often with one community containing almost all the nodes and a few very small communities. This imbalance is incompatible with our approach, which requires strict control over the size of the subgraphs so that their embedding fits in GPU memory. Moreover, some of the communities are disconnected. On the two largest graphs, YAGO4 + taxonomy and Freebase, Louvain exceeds the time limit ($2 \cdot 10^4$ seconds).

**Leiden** modifies Louvain to guarantee connected communities and more stable outputs. It has very good scaling capabilities (Figure 11) but shares with Louvain the issue of producing highly-imbalanced communities.

**LE** is a recursive algorithm that splits nodes based on the sign of their corresponding coefficient in the leading eigenvector of the modularity matrix. If these signs are all the same, the algorithm does not split the network further. Experimentally, LE returns only one partition (containing all the nodes) for YAGO3, YAGO4, YAGO4 + taxonomy, and Freebase. For YAGO4.5 + taxonomy, it hits our pre-set time limit ($2 \cdot 10^4$ seconds). Therefore, we only report its performance for Mini YAGO3 and YAGO4.5, for which it outputs 2 and 4 partitions, respectively. It is important to note that the more communities it returns, the longer it takes to run because the algorithm proceeds recursively.

**Infomap** uses random walks and information theory to group nodes into communities. Experimentally, it produces a lot of small communities with no connectedness guarantee. Additionally, it is too slow to be used on large graphs.

**LPA** propagates labels across the network iteratively, allowing densely connected nodes to form communities. Similarly to Louvain and Leiden, the downside is that it does not control the size of the detected communities. It is also too slow to run on the largest graphs.

**SC** uses the smallest eigenvectors of the graph Laplacian to transform the graph into a low-dimensional space and then applies k-means to group nodes together. However, the expensive eigenvector computation is a bottleneck that does not allow this approach to be used on huge graphs.

**HDRF** is a streaming algorithm that produces balanced edge partitions (vertex cut) and minimizes the replication factor. However, it produces disconnected partitions, and it is too slow to run on the largest graphs.

Therefore, none of the above methods readily produces subgraphs suitable for SEPAL. Indeed Figure 12 shows that BLOCS is the only method that returns subgraphs that are both connected and bounded in size, while being competitive in terms of scalability (Figure 11).

**BLOCS cannot be replaced with METIS**  To demonstrate the benefits of BLOCS over existing methods, we try to replace BLOCS with METIS in our framework. The results are presented in Figure 14.

Two important points differentiate these methods:

1. Contrary to BLOCS, METIS outputs disconnected partitions (see Figure 12). Given the structure of SEPAL, this results in zero-embeddings for entities not belonging to the core connected component at propagation time. Interestingly, the presence of zero-embeddings affects downstream scores very little, likely because most downstream entities belong to the core connected component and are thus not impacted by this.

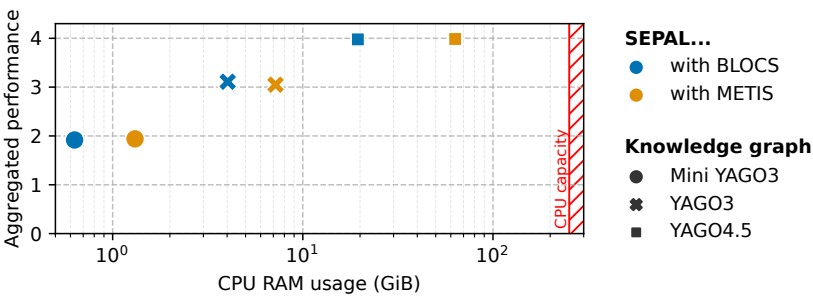

Figure 14: **Ablation study: replacing BLOCS with METIS.** Normalized R2 scores (same as Figure 6) aggregated across evaluation datasets (movie revenues, US accidents, US elections, housing prices) for SEPAL with BLOCS and METIS are plotted against CPU RAM usage. BLOCS necessitates significantly less memory than METIS. We were not able to run SEPAL + METIS on knowledge graphs larger than YAGO4.5, hitting CPU RAM limits during the partitioning stage.

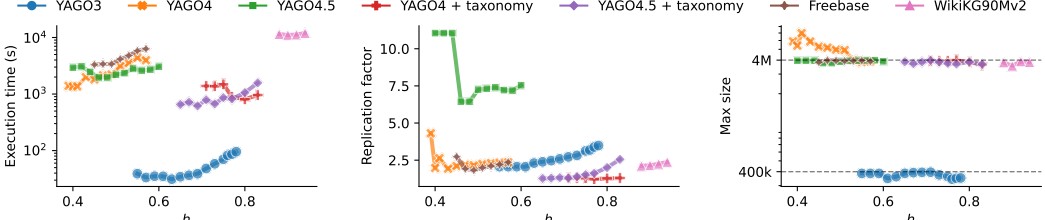

Figure 15: **Sensitivity analysis to parameter** $h$. Effect of varying $h$ on BLOCS' execution time, replication factor (average number of outer subgraphs containing a given entity), and maximum subgraph size. A lower bound of the domain of $h$ values to explore for a dataset is given by the proportion of nodes that are neighbors to super-spreaders. Indeed, as BLOCS' first step is to assign super-spreaders neighbors, if $h$ is smaller than this value, BLOCS completely skips the diffusion phase. For YAGO4.5 and YAGO4, this results in sharp variations of the replication factor or the maximum subgraph size, respectively. The maximum size parameter $m$ was set to 400k for YAGO3, and 4M for others.

2. METIS does not scale as well as BLOCS in terms of CPU memory. On our hardware, SEPAL + METIS could not scale to graphs larger than YAGO4.5 (32M entities), and therefore, BLOCS is indispensable for very large knowledge graphs.

### E.4 Effect of BLOCS' stopping diffusion threshold

In the BLOCS algorithm, the hyperparameter $h$ controls the moment of the switch from diffusion to dilation. For $h \in (0, 1)$, BLOCS stops diffusion once the proportion of entities of the graph assigned to a subgraph is greater than or equal to $h$.

Figure 15 shows that increasing $h$ tends to increase the execution time and the overlap between subgraphs. Higher overlaps can be preferable to enable the information to travel between outer subgraphs during embedding propagation. However, a high overlap also incurs additional communication costs because the embeddings are moved several times from CPU to GPU, increasing SEPAL's overall execution time.

### E.5 How distant from the core are the outer entities?

Figure 16 shows that outer entities are in average very close to the core subgraph. This is due to the fact that the core contains the most central entities, and to the scale-free nature of knowledge graphs.

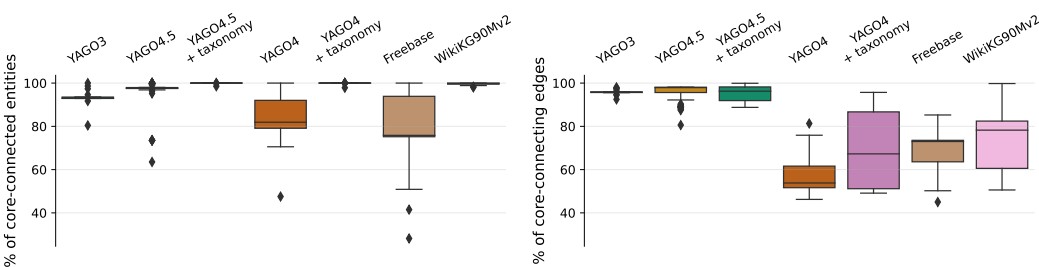

Figure 16: **Outer subgraphs are well connected to the core.** The left plot gives the percentage of outer entities that are directly connected to the core subgraph. The right plot gives the percentage of edges that come from the core subgraph among all the edges coming to a given outer subgraph, showing the amount of information transferred from the core to outer subgraphs relative to outer-outer communication.

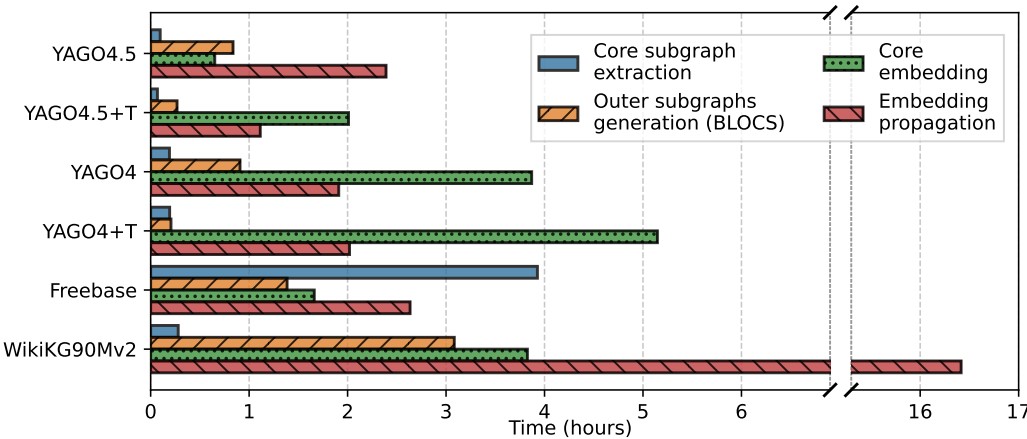

Figure 17: **SEPAL's execution time breakdown.** Execution time of SEPAL's different components, for the best-performing configurations of SEPAL on downstream tasks (see Table 9 for hyperparameters values).

# F   Further analysis of SEPAL

## F.1   Execution time breakdown

Here, we present the contribution of each part of the pipeline to the total execution time. Specifically, we break down our method into four parts:

1. core subgraph extraction;
2. outer subgraphs generation (BLOCS);
3. core embedding;
4. embedding propagation.

Figure 17 shows the execution time of the different components of SEPAL. It includes six large-scale knowledge graphs, for which the execution times have the same order of magnitude. The results reveal that most of the execution time is due to the core embedding and embedding propagation phases, while the core extraction time is negligible.

Four key factors influence SEPAL's execution time during the four main steps of the pipeline:

1. **The core selection strategy**: the degree-based selection is faster than the hybrid selection. For the hybrid selection, the factor that influences the speed the most is the number of distinct relations.

2. **The diameter of the knowledge graph**: graphs with large diameters call for more dilation steps during BLOCS' subgraph generation, and dilation is more costly than diffusion because it requires checking node assignments. This explains why adding the taxonomies to YAGO4 and YAGO4.5 drastically reduces the time required to run BLOCS, as shown on Figure 17.

3. **The core subgraph size**: the more triples in the core subgraph, the longer the core embedding. This explains the wide disparities between the core embedding times on Figure 17, despite all the core subgraphs having roughly the same number of entities: YAGO4 core subgraph is more dense (33M triples), compared to YAGO4.5 (7M triples), for instance. The core embedding time also depends on hyperparameters such as the number of training epochs.

4. **The total number $N$ of entities in the graph**: this number determines the size of the embedding matrix. The communication cost of moving embedding matrices from CPU to GPU, and vice versa, accounts for most of the propagation time, and increases with $N$. It also increases with the amount of overlap between the outer subgraphs produced by BLOCS,

explaining the differences in propagation time between YAGO4.5 and YAGO4.5 + taxonomy for instance.

The number of propagation steps $T$ has little impact on the embedding propagation time. The reason for this is that much of this time stems from the communication cost of loading the embeddings onto the GPU, and not from performing the propagation itself.

## F.2 SEPAL's hyperparameters

### F.2.1 List of SEPAL's hyperparameters

Here, we list the hyperparameters for SEPAL, and discuss how they can be set. Table 9 gives the values of those that depend on the dataset.

- **Proportion of core nodes** $\eta_n$: the idea is to select it large enough to ensure good core embeddings, but not too large so that core embeddings fit in the GPU memory. Figure 18 shows the experimental effect of varying this parameter;

- **Proportion of core edges** $\eta_e$: increasing it at the expense of $\eta_n$ (to keep the core size within GPU memory limits) improves the relational coverage, but reduces the density of the core. Sparser core subgraphs tend to deteriorate the quality of SEPAL's embeddings for feature enrichment (Figure 22). However a good relational coverage is essential for better link-prediction performance;

- **Stopping diffusion threshold** $h$: it depends on the graph structure, and tuning is done empirically by monitoring the proportion of unassigned entities during the BLOCS algorithm: $h$ is chosen equal to the proportion of assigned entities at which BLOCS starts to stagnate during its diffusion regime. In practice, Figure 15 and Figure 19 shows that as long as $h$ is chosen greater than the proportion of entities that are neighbors of super-spreaders, the algorithm is not too sensitive to this parameter (otherwise, it skips the diffusion phase, which can be detrimental);

- **Number of propagation steps** $T$: it is chosen high enough to ensure reaching the remote entities (otherwise, they will have zeros as embeddings). Taking $T$ equal to the graph's diameter guarantees that this condition is fulfilled. However, for graphs with long chains, this may slow down SEPAL too much. In practice, setting $T$ at 2–3 times the Mean Shortest Path Length (MSPL) usually embeds most entities effectively;

- **Propagation learning rate** $\alpha$: it controls the proportion of self-information relatively to neighbor-information during propagation updates. For the DistMult model, this parameter has no effect during the first propagation step, when an entity is reached for the first time because outer embeddings are initialized with zeros and the neighbors' message is normalized. In practice, the embedding of an outer entity can reach in one step a position very close to its fix point, and thus this parameter does not have much effect (Figure 20). For our experiments we typically use $\alpha = 1$;

- **Subgraph maximum size** $m$: the idea is to use the largest value for which it is possible to fit the subgraph's embeddings in the GPU memory. We use $4 \cdot 10^4$ for Mini YAGO3, $4 \cdot 10^5$ for YAGO3, $2 \cdot 10^6$ for WikiKG90Mv2, and $4 \cdot 10^6$ for the other knowledge graphs;

- **Embedding dimension** $d$: we use $d = 100$ (except for complex embeddings, where $d = 50$ to keep the same number of parameters);

- **Number of epochs for core training** $n_{\text{epoch}}$: see Table 9;

- **Batch size for core training** $b$: see Table 9;

- **Optimizer for core training**: we use the Adam optimizer with learning rate $\text{lr} = 1 \cdot 10^{-3}$;

- **Number $p$ of negative samples per positive for core training**: we use $p = 100$ (Table 9).

### F.2.2 Experimental study of hyperparameter effect

Figure 18, Figure 19, Figure 20 and Figure 21 investigate the sensitivity of SEPAL to different hyperparameters. The hyperparameter that seems to impact the most SEPAL's performance is the core proportion $\eta_n$. Indeed, Figure 18 shows that increasing $\eta_n$ tends to improve embedding quality

| Dataset | Parameter | Grid (best in bold) |
|---|---|---|
| Mini YAGO3 | $\eta_n/\eta_e$ | 0.05/—, 0.2/—, 0.025/0.005, **0.3/0.05** |
| | $h$ | 0.6, 0.65, 0.7, 0.75, **0.8** |
| | $T$ | 2, **5**, 10, 15, 25 |
| | $n_{\text{epoch}}$ | 12, 25, 50, 60, **75** |
| | $b$ | **512** |
| | $p$ | 1, **100**, 1000 |
| YAGO3 | $\eta_n/\eta_e$ | 0.05/—, **0.1/—**, 0.025/0.015, 0.3/0.05 |
| | $h$ | 0.55, 0.6, 0.65, 0.7, 0.75, **0.77** |
| | $T$ | 2, 5, 10, **15**, 25 |
| | $n_{\text{epoch}}$ | 16, **18**, 25, 45, 50, 75 |
| | $b$ | **2048**, 4096, 8192 |
| | $p$ | 1, **100**, 1000 |
| YAGO4.5 | $\eta_n/\eta_e$ | 0.03/—, **0.015/0.01**, 0.05/0.03 |
| | $h$ | 0.4, 0.5, **0.6** |
| | $T$ | 2, 5, 10, 15, 25, **50** |
| | $n_{\text{epoch}}$ | 16, **24**, 75 |
| | $b$ | **8192**, 16384 |
| | $p$ | 1, **100** |
| YAGO4.5+T | $\eta_n/\eta_e$ | **0.03/—**, 0.015/0.005, 0.04/0.015 |
| | $h$ | **0.8** |
| | $T$ | 2, 5, 10, 15, **20**, 25 |
| | $n_{\text{epoch}}$ | **32**, 75 |
| | $b$ | **8192** |
| | $p$ | **100** |
| YAGO4 | $\eta_n/\eta_e$ | 0.03/—, **0.015/0.005**, 0.012/0.025 |
| | $h$ | **0.55** |
| | $T$ | 2, 5, 10, 15, **20**, 25 |
| | $n_{\text{epoch}}$ | 4, **28**, 75 |
| | $b$ | **8192**, 65536 |
| | $p$ | 1, **100** |
| YAGO4+T | $\eta_n/\eta_e$ | 0.02/—, **0.01/0.005** |
| | $h$ | 0.45, **0.8** |
| | $T$ | 10, **20** |
| | $n_{\text{epoch}}$ | **32**, 75 |
| | $b$ | **8192**, 65536 |
| | $p$ | 1, **100** |
| Freebase | $\eta_n/\eta_e$ | 0.02/—, **0.01/0.005** |
| | $h$ | **0.55** |
| | $T$ | **15** |
| | $n_{\text{epoch}}$ | **24** |
| | $b$ | **8192** |
| | $p$ | **100** |
| WikiKG90Mv2 | $\eta_n/\eta_e$ | **0.02/—** |
| | $h$ | **0.92** |
| | $T$ | **10** |
| | $n_{\text{epoch}}$ | **12** |
| | $b$ | **8192** |
| | $p$ | **100** |

Table 9: **Hyperparameter search space** for SEPAL, and best values (in bold) for each knowledge graph on downstream tasks. The best values are those that gave the best average performance on the 4 validation tables and that were used to get the results in Figure 2 and Figure 3.

for downstream tasks. However, the effect seems to be plateauing relatively fast for YAGO3 (not much improvement between $\eta_n = 5\%$ and $\eta_n = 10\%$). For other datasets (YAGO4.5, YAGO4), it is not possible to explore larger values of $\eta_n$ because the core subgraph would not fit in the GPU memory. Moreover, decreasing $\eta_n$ makes SEPAL run faster, as the core embedding phase accounts for a substantial share of the total execution time (Figure 17). There is, therefore, a trade-off between time and performance.

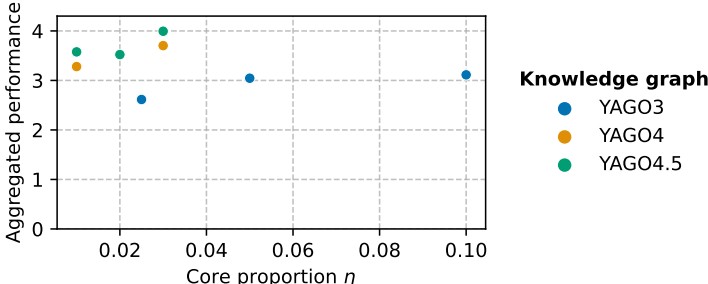

Figure 18: Effect of core proportion $\eta_n$ on SEPAL's performance, with the degree-based core selection strategy.

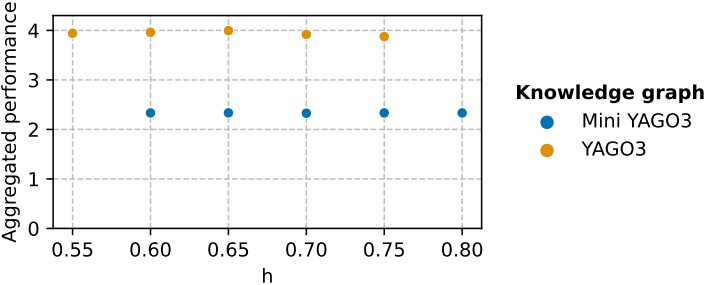

Figure 19: Effect of stopping diffusion threshold $h$ on SEPAL's performance.

**Core selection strategy** *Degree-based selection:* The simple degree-based core selection strategy is convenient for two reasons:

1. Degree is inexpensive to compute, ensuring the core extraction phase to be fast (see Figure 17);

2. It yields very dense core subgraphs. Indeed, while they contain $\eta_n\%$ of the entities of the full graph, they gather around $4\eta_n\%$ of all the triples (Table 10). This allows the training on the core to process a substantial portion of the knowledge-graph triples, resulting in richer representations.

*Hybrid selection:* However, a problem with the degree-based selection is that some relation types may not be included in the core subgraph. To deal with this issue, SEPAL also proposes a more complex *hybrid* method for selecting the core subgraph, that incorporates the relations. It proceeds in four main steps:

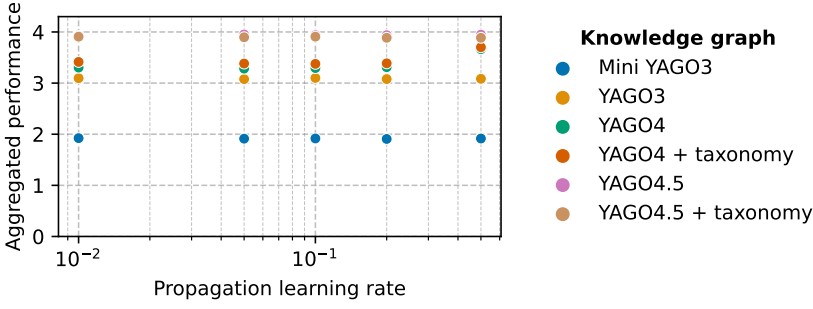

Figure 20: Effect of propagation learning rate $\alpha$ on SEPAL's downstream performance.

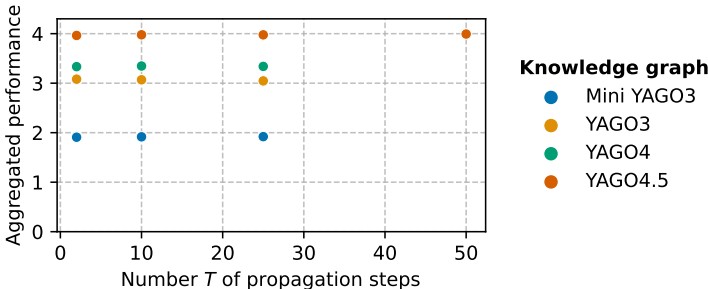

Figure 21: Effect of number $T$ of propagation steps on SEPAL's downstream performance.

Table 10: **Effect of core selection strategies.** Number of entities and triples inside the core subgraph and proportion of the full graph they represent (in parentheses). $\eta_n$ and $\eta_e$ are the hyperparameters for nodes and edges, respectively. Column *#Rel* gives the number of relation types present in the core compared to the total number of relation types in the knowledge graph. We highlight in red the cases where some relations are missing. Column *Time* gives the measured computation time for core selection.

| | **Strategy** | $\boldsymbol{\eta_n}$ | $\boldsymbol{\eta_e}$ | **#Rel** | **#Entities** | **#Triples** | **Time** |
|---|---|---|---|---|---|---|---|
| YAGO3 | Degree | 5% | - | 37/37 | 126k (4.9%) | 1.0M (18.5%) | 17s |
| | Hybrid | 2.5% | 1.5% | 37/37 | 121k (4.7%) | 733k (13.1%) | 20s |
| YAGO4.5 | Degree | 3% | - | 62/62 | 932k (2.9%) | 7.2M (9.6%) | 2min |
| | Hybrid | 1.5% | 1% | 62/62 | 1.1M (3.3%) | 5.7M (7.5%) | 6min |
| YAGO4 | Degree | 3% | - | 61/76 | 1.1M (3.0%) | 33M (13.4%) | 8min |
| | Hybrid | 1.5% | 0.5% | 76/76 | 1.4M (3.8%) | 28M (11.1%) | 11min |
| YAGO4.5+T | Degree | 3% | - | 64/64 | 1.5M (3.0%) | 13M (9.9%) | 4min |
| | Hybrid | 1.5% | 0.5% | 64/64 | 1.2M (2.5%) | 8.3M (6.5%) | 5min |
| YAGO4+T | Degree | 2% | - | 64/78 | 1.3M (2.0%) | 41M (12.8%) | 9min |
| | Hybrid | 1% | 0.5% | 78/78 | 1.5M (2.3%) | 32M (10.1%) | 12min |
| Freebase | Degree | 2% | - | 5,363/14,665 | 1.7M (2.0%) | 15M (4.4%) | 9min |
| | Hybrid | 1% | 0.5% | 14,665/14,665 | 1.9M (2.3%) | 14M (4.1%) | 4h |
| WikiKG90Mv2 | Degree | 2% | - | 886/1,387 | 1.8M (2.0%) | 62M (10.4%) | 17min |
| | Hybrid | 0.4% | 0.7% | 1,387/1,387 | 2.0M (2.2%) | 37M (6.2%) | 48min |

1. **Degree selection**: Sample the nodes with the top $\eta_n$ degrees.

2. **Relation selection**: Sample the edges with the top $\eta_e$ degrees (sum of degrees of head and tail) for each relation type, and keep the corresponding entities.

3. **Merge**: Take the union of these two sets of entities.

4. **Reconnect**: If the induced subgraph has several connected components, add entities to make it connected. This is done using a breadth-first search (BFS) with early stopping from the node with the highest degree of each given connected component (except the largest) to the largest connected component. For each connected component (except the largest), a path linking it to the largest connected component is added to the core subgraph.

This way, each relation type is guaranteed to belong to the core subgraph, by design. Table 10 confirms this experimentally, even for Freebase and its 14,665 relation types. This method features two hyperparameters $\eta_n$ and $\eta_e$, the proportions for node and edge selections, controlling the size of its output subgraph. The values we used are provided in Table 10 for each dataset.

Regarding performance, Figure 22 shows that the hybrid strategy slightly underperforms the degree-based approach on four real-world downstream tasks. Indeed, the hybrid approach enhances relational coverage but, as a counterpart, yields a sparser core subgraph (see *#Triples* in Table 10), which is detrimental to downstream performance. However, one can adjust the values of $\eta_n$ and $\eta_e$ to control the trade-off between downstream performance and relation coverage, depending on the use case.

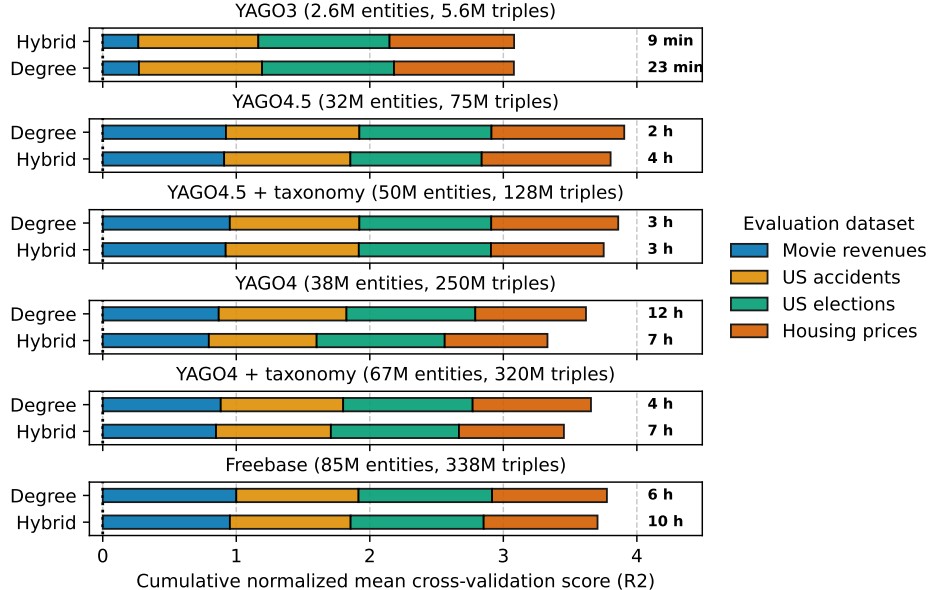

Figure 22: Performance of SEPAL+DistMult for the two different core selection strategies. We use the hyperparameters of Table 10. The simpler degree-based selection strategy runs faster and performs better on downstream tasks.

For instance, for the link prediction task, a better relational coverage (greater $\eta_e$) improves the performance (Appendix C.6).

**Exploring other centrality measures**   Here, we compare degree with PageRank as a centrality measure for the core selection (keeping all other hyperparameters unchanged). Like the degree, the PageRank can be computed very efficiently on huge graphs using sparse matrix multiplications.

Table 11 reports results on the real-world downstream tasks. Regarding performance, it seems that the difference between degree and PageRank depends on the graph's structure. Specifically, we observe that for YAGO4, YAGO4+T, Freebase, and WikiKG90Mv2, PageRank improves the performance. This echoes the results of Figure 16, which show that in these four specific KGs, contrary to the others, the degree-based core selection yields cores that are connected through fewer edges to some outer subgraphs. We can therefore conjecture that for these KGs, PageRank improves information flow and mitigates issues like oversquashing during propagation, ultimately increasing performance. Regarding computational cost, we see that SEPAL with PageRank usually runs slightly faster than with degree. This is because PageRank yields slightly sparser cores, leading to faster core training.

Table 11: **Effect of core selection strategy: PageRank vs Degree.** We compare the performance of SEPAL+DistMult on real-world downstream tasks when using either degree or PageRank as a centrality measure for core selection. We report the average normalized R2 score across the four downstream tasks, along with the total execution time (in parentheses).

| Dataset | Core defined by Degree | Core defined by PageRank |
|---|---|---|
| YAGO3 | **0.783** (11m 20s) | 0.742 (9m 23s) |
| YAGO4 | 0.817 (6h 20m) | **0.861** (4h 53m) |
| YAGO4+T | 0.815 (10h 9m) | **0.881** (8h 16m) |
| YAGO4.5 | **0.949** (4h 11m) | 0.925 (4h 3m) |
| YAGO4.5+T | **0.923** (2h 47m) | 0.912 (2h 40m) |
| Freebase | 0.917 (5h 58m) | **0.919** (5h 58m) |
| WikiKG90Mv2 | 0.898 (20h 31m) | **0.902** (23h 59m) |

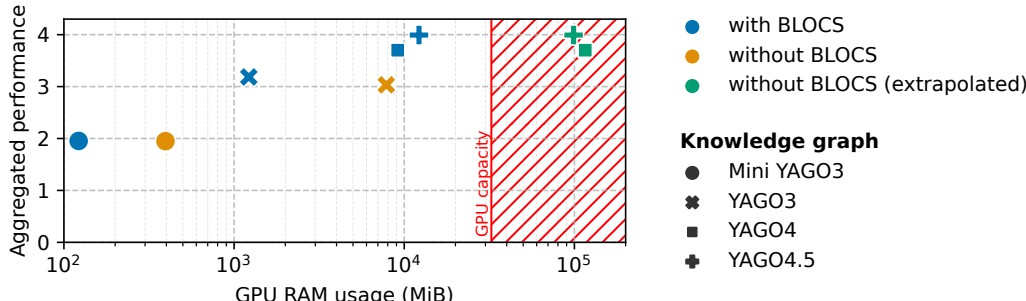

Figure 23: **Ablation study: BLOCS scales SEPAL memory-wise.** Normalized R2 scores aggregated across evaluation datasets (movie revenues, US accidents, US elections, housing prices) for SEPAL with and without BLOCS are plotted against GPU RAM usage. BLOCS preserves performance for a given knowledge graph while drastically reducing memory pressure on GPU RAM. Without BLOCS, the GPU runs out of memory for YAGO4 and YAGO4.5.

### F.3 Ablation study: SEPAL without BLOCS

Here, we study the effect of removing BLOCS from our proposed method. On smaller knowledge graphs, SEPAL can be used with a simple core subgraph extraction and embedding followed by the embedding propagation. This ablation reveals the impact of BLOCS on the model's performance. Figure 23 shows that adding BLOCS to the pipeline on graphs that would not need it (because they are small enough for all the embeddings to fit in GPU memory) does not alter performance. Additionally, BLOCS brings scalability. By tuning the maximum subgraph size $m$ hyperparameter, one can move the blue points horizontally on Figure 23 and choose a value within the GPU memory constraints. There is a trade-off between decreasing GPU RAM usage (*i.e.*, moving the blue points to the left) and increasing execution time, as fewer entities are processed at the same time.

### F.4 Speedup over base embedding model

Figure 24 demonstrates that SEPAL can accelerate its base embedding model by more than a factor of 20, while also boosting its performance on downstream tasks.

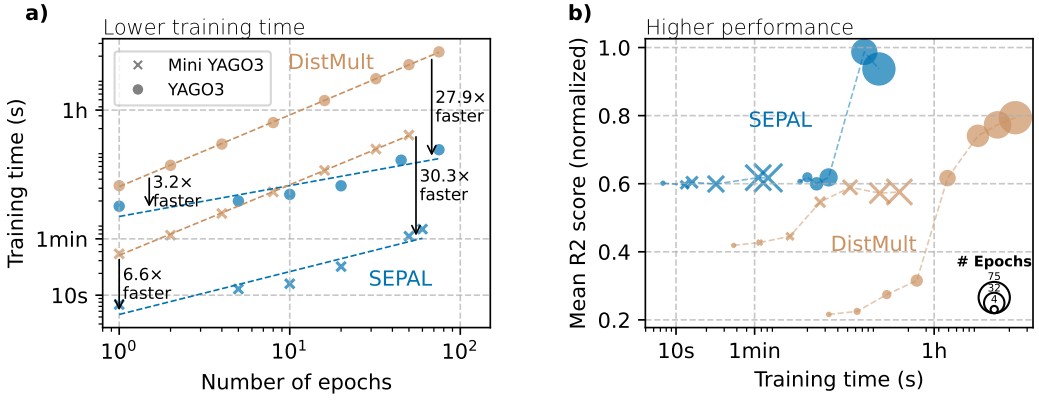

Figure 24: **Comparison between SEPAL and its base embedding model. a) Computation time per training iteration.** For a given training configuration, SEPAL is up to 30 times faster than its base embedding algorithm DistMult. **b) Learning curves.** SEPAL achieves strong downstream performance much quicker than DistMult. For both plots, we only vary the number of epochs and fix the following parameters for DistMult's training and SEPAL's core training: $p = 1$, lr $= 1 \cdot 10^{-3}$, $b = 512$ for Mini YAGO3 and $b = 2048$ for YAGO3. For SEPAL, we use the degree-based core selection with $\eta_n = 5\%$.

Table 12: **Comparison between SEPAL and DistMult-ERAvg** on Mini YAGO3. We report the normalized mean cross-validation score (R2) across the four real-world downstream tasks, along with the total training time. SEPAL outperforms DistMult-ERAvg while being significantly faster.

| Method | Housing prices | Movie revenues | US accidents | US elections | Time |
|---|---|---|---|---|---|
| DistMult-ERAvg | 0.149 | 0.124 | 0.444 | 0.916 | 2h 41m |
| SEPAL | **0.276** | **0.159** | **0.548** | **0.929** | **5m** |

Moreover, the speedup increases with the number of training epochs, as SEPAL's constant-cost steps (core extraction, BLOCS, and propagation) are amortized when core training gets longer. Indeed, each additional epoch is cheaper with SEPAL than with DistMult, since training is done only on the smaller core rather than the full graph.

# G   Discussion

## G.1   Comparison to prior work

### G.1.1   Comparison to DistMult-ERAvg

Albooyeh et al. [2020] propose an aggregation that is similar to SEPAL's aggregation during the propagation phase, however the two methods optimize the embeddings in two very distinct ways:

- Albooyeh et al. [2020] introduce propagation *within* the standard link prediction pipeline: during training, for each triple $(v, r, u)$, they occasionally (with probability $p$) replace one entity's embedding with an aggregated version (e.g., $\theta_u \cdot \theta_r$) before computing the plausibility score. However, training still relies on negative sampling and a classic link prediction loss, and thus optimizes for local triple-level contrasts like traditional KGE methods.
- SEPAL, in contrast, explicitly separates the optimization objective: embeddings for a small core are trained with a classic KGE objective, and then relation-aware propagation is used across the rest of the graph, without negative sampling. This distinction is crucial: SEPAL removes the need for negative sampling on typically 95 to 99% of the graph, enabling both improved scalability and alignment properties beneficial for downstream tasks.

Moreover, DistMult-ERAvg is not designed for very large graphs. On the datasets considered in this paper, it fails with out-of-memory errors on all the graphs except Mini YAGO3 (129k entities, 1.1M triples).

Table 12 reports performance and runtime on Mini YAGO3, showing that SEPAL is 32 times faster than DistMult-ERAvg while achieving consistently better scores. This is expected since DistMult-ERAvg follows the classic optimization loop with negative sampling, gradient computations, and parameter updates, and therefore inherits the limitations of traditional KGE methods. The strength of DistMult-ERAvg lies in out-of-sample embedding computation, which SEPAL also supports (see Appendix G.3), but with greater scalability.

### G.1.2   Comparison to NodePiece

SEPAL shares with NodePiece the fact that it embeds a subset of entities. Parallels can be drawn between: a) the anchors of NodePiece and the core entities of SEPAL; b) the encoder function of NodePiece and the embedding propagation of SEPAL. Yet, our approach differs from NodePiece in several ways.

**Neighborhood context handling**   Both methods handle completely differently the neighborhood of entities. NodePiece tokenizes each node into a sequence of $k$ anchors and $m$ relation types, where $k$ and $m$ are fixed hyperparameters shared by all nodes. If the node degree is greater than $m$, NodePiece downsamples randomly the relation tokens, and if it is lower than $m$, [PAD] tokens are appended; both seem sub-optimal. In contrast, SEPAL accommodates any node degree and uses all the neighborhood information, thanks to the message-passing approach that handles the neighborhood context naturally.

Additionally, NodePiece's tokenization relies on an expensive BFS anchor search, unsuitable for huge graphs. On our hardware, we could not run the vanilla NodePiece (PyKEEN implementation)

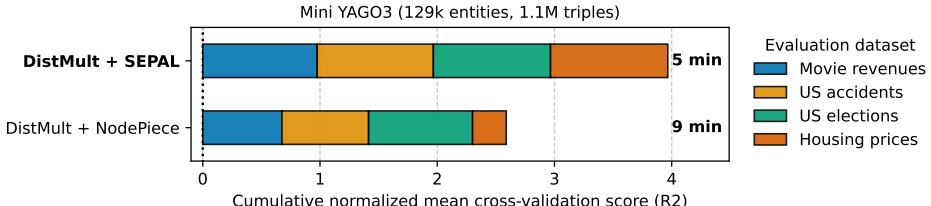

Figure 25: Comparing SEPAL with NodePiece on Mini YAGO3.

on graphs bigger than Mini YAGO3 (129k entities). For YAGO3 and YAGO4.5, we had to run an ablated version where nodes are tokenized only from their relational context (i.e., $k = 0$, studied in the NodePiece paper with good results), to skip the anchor search step.

**Training procedure**  At train time, NodePiece goes through the full set of triples at each epoch to optimize both the anchors' embeddings and the encoder function parameters, necessitating many gradient computes and resulting in long training times for large graphs. On the contrary, SEPAL performs mini-batch gradient descent only on the triples of the core subgraph, which provides significant time savings. To illustrate this, Figure 25 compares the performance of SEPAL and vanilla NodePiece on Mini YAGO3, showing that SEPAL outperforms NodePiece on downstream tasks while being nearly two times quicker.

**Embedding propagation to non-anchor/non-core entities**  To propagate to non-anchor entities, NodePiece uses an encoder function (MLP or Transformer) that has no prior knowledge of the relational structure of the embedding space, and has to learn it through gradient descent. On the contrary, SEPAL leverages the model-specific relational structure to compute the outer embeddings with no further training needed.

### G.2 Communication costs of SEPAL

**SEPAL optimizes data movement**  In modern computing architectures, memory transfer costs are high, amounting to much computation, and the key to achieve high operation efficiency is to reduce data movement [Mutlu et al., 2022].

For distributed methods such as PBG or DGL-KE, parallelization incurs additional communication costs due to two factors: *1)* Learned parameters shared across workers (relation embeddings) require frequent synchronization *2)* Entities occurring in several triples belonging to different buckets have their embeddings moved several times from CPU to GPU, and this for every epoch. This latter effect can be mitigated by better partitioning, but remains significant.

SEPAL avoids much of the communication costs by keeping the core embeddings on GPU memory throughout the process. These core embeddings are the ones that would move the most in distributed settings, because they correspond to high-degree entities involved in many triples. Moreover, SEPAL's propagation loads each outer subgraph only once on the GPU, contrary to other methods that perform several epochs. This significantly reduces data movement for outer embeddings: empirically, they only cross the CPU/GPU boundary twice on average (see Table 13).

**Estimating I/O**  Communication costs occur in SEPAL during the propagation phase, where the embeddings of each of the subgraphs generated by BLOCS have to be loaded on the GPU, subgraph after subgraph. We analyze the number $x$ of back-and-forth of a given embedding between CPU and GPU memory. The optimal value is $x = 1$, meaning the embeddings are transferred only once. A detailed breakdown follows:

Table 13: Empirical average number of memory transfers between CPU and GPU for outer embeddings across datasets.

| Dataset | $x_{\text{outer}}$ |
|---|---|
| Mini YAGO3 | 1.20 |
| YAGO3 | 3.07 |
| YAGO4.5 | 7.47 |
| YAGO4.5+T | 1.94 |
| YAGO4 | 2.29 |
| YAGO4+T | 1.27 |
| Freebase | 2.09 |
| WikiKG90Mv2 | 1.84 |
| **Average** | **2.65** |

- **For core embeddings:** core embeddings remain at all times on the GPU memory. They are only moved to the CPU once at the end, to be saved on disk with the rest of the embeddings. Therefore, the data movement of core embeddings is optimal: $x_{\mathrm{core}} = 1$.
- **For outer embeddings:** SEPAL loads each outer subgraph only once to the GPU. Consequently, the average number $x_{\mathrm{outer}}$ of memory transfers for outer embeddings corresponds to the average number of subgraphs in which a given outer entity appears. Table 13 reports empirical values of $x_{\mathrm{outer}}$ across datasets, ranging from 1.20 to 7.47, with an overall average of 2.65. This redundancy arises from subgraph overlap, which is directly influenced by the $h$ hyperparameter in BLOCS: smaller values of $h$ yield fewer diffusion steps (and more dilation steps), leading to reduced subgraph overlap and, hence, fewer memory transfers per embedding.

The optimal value of $x = 1$ can only be achieved in the case where the entire graph fits in GPU memory. Every approach that scales beyond GPU RAM limits has its communication overheads, i.e., $x > 1$.

For comparison, distributed training schemes that load buckets of triples to GPU iteratively exhibit significantly higher communication costs. In such settings, the number of memory transfers for the embedding of an entity $u$ is given by $x = n_{\mathrm{buckets}}(u) \times n_{\mathrm{epochs}}$ where $n_{\mathrm{buckets}}(u)$ is the number of buckets that contain a triple featuring entity $u$, and $n_{\mathrm{epochs}}$ is the number of epochs. That is at least one or two orders of magnitude greater than what SEPAL achieves in terms of data movement. Indeed, knowledge-graph embedding methods are usually trained for a few tens if not a few hundreds of epochs, so $x$ is much bigger than 10, not even taking into account $n_{\mathrm{buckets}}(u)$ that can be large, depending on the graph structure and on the quality of the graph partitioning.

### G.3 Outlook on continual learning

The modular nature of SEPAL makes it well-suited for continual learning scenarios, where new entities are added to the knowledge graph over time. Indeed, new embeddings can be computed without retraining from scratch, via a few additional propagation steps, as long as the relations remain unchanged. To demonstrate this, we use two versions of the YAGO3 knowledge graph: the original 2014 release, and the 2022 revived version. Table 14 gives the statistics of these two datasets.

Table 14: Statistics of the two versions of YAGO3 used to illustrate SEPAL's suitability for continual learning.

| Version | #Entities | #Relations | #Triples |
|---|---|---|---|
| YAGO3-2014 | 2,570,716 | 37 | 5,585,004 |
| YAGO3-2022 | 4,546,966 | 37 | 14,691,781 |

We adapt SEPAL to this continual learning setting by: (1) initializing the embeddings of YAGO3-2022 using embeddings precomputed on YAGO3-2014, (2) propagating embeddings for 5 additional steps to update existing nodes and embed new entities. We denote this method SEPAL-CL in the results.

Table 15 shows that:

- For downstream applications, YAGO3-2022 brings value compared to YAGO3-2014. Indeed, for all the methods considered, the embeddings learned on the 2022 dataset score higher than the strongest method on YAGO3-2014, SEPAL.
- SEPAL trained from scratch is $10 \times$ faster than DistMult on YAGO3-2022.
- SEPAL-CL is $57 \times$ faster than SEPAL trained from scratch on YAGO3-2022, and $587 \times$ faster than DistMult.

Table 15: **Continual learning experiment on YAGO3.** Average normalized R2 scores across the four real-world downstream regression tasks (movie revenues, US accidents, US elections, housing prices) and total runtime for different methods. SEPAL-CL denotes SEPAL in the continual learning setting, where new entities are embedded via propagation.

| YAGO3 version | Method | Average performance | Runtime |
|---|---|---|---|
| 2014 | SEPAL | 0.836 | 0h 11m 20s |
| 2022 | DistMult | 0.884 | 11h 05m 36s |
| 2022 | SEPAL | 0.988 | 1h 05m 07s |
| 2022 | SEPAL-CL | 0.962 | 0h 01m 08s |

- SEPAL-CL clearly outperforms DistMult, and almost matches the performance of SEPAL trained from scratch, even in this very challenging scenario (8 years between the two versions of the graph), where the size of the graph has doubled in terms of entities, and tripled in terms of triples. In a real-life application, we could imagine embeddings recomputed monthly at very low cost.

## G.4  Broader impacts

SEPAL may reflect the biases present in the training data. For instance, Wikipedia, from which YAGO is derived, under-represents women [Reagle and Rhue, 2011]. We did not evaluate how much our method captures such biases. We note that the abstract nature of embeddings may make the biases less apparent to the user; however, this problem is related to embeddings and not specific to our method. It may be addressed by debiasing techniques [Bolukbasi et al., 2016, Fisher et al., 2020] for which SEPAL could be adapted.

