# OpenReview forum: "Scalable Feature Learning on Huge Knowledge Graphs for Downstream Machine Learning"
_NeurIPS.cc/2025/Conference — NeurIPS 2025 poster_

### Official Review · Reviewer_UiQt · 2025-06-30

**Clarity:** 2
**Significance:** 2
**Originality:** 2
**Rating:** 3
**Confidence:** 4

**Summary:**

Proposes a method to obtain knowledge graph embeddings for large knowledge graphs. This is done by decomposing the graph into a "core" (used for training and to obtain embeddings of frequent entities) and multiple potentially overlapping "rest" subgraphs (updated via message passing to obtain embeddings of remaining entities). Reports on an experimental study.

**Questions:**

/

**Ethical Concerns:**

["NO or VERY MINOR ethics concerns only"]

**Final Justification:**

The authors conducted a number of additional experiments in their final response, which strengthens their case. I feel that this paper is not ready, though, but requires a revision (which we do not have here, unfortunately).

> On W1. My key point is that scalable (and also low-cost / single-GPU) methods exist but have not been considered in the experimental study.

The authors report numbers on SMORE, an older method, in their response and state that they could not get Marius* to run. This addresses this point somewhat, and could be fully addressed by rewriting the paper and removing claims that existing methods "struggle to scale", as they clearly do not.

> On W2-1) This objective is not well-motivated then, and I did not fully understand the alignment discussion in the paper. I found the response somewhat confusing because this objective is not used for the core (why not?), and quality ("alignment") and scalability (core+propagation) arguments are mixed in the paper.

This point is not addressed (as it would require revising that part), but the additional experimental results provided by the authors are promising and indicate that the "alignment" may be useful.

> On W2-2) A comparison to this work is missing in the paper (both in words and in experimental study).

Here the quick experimental comparison again looks favorable for the proposed method. A more thorough setup and discussion of technical differences is still open though; that's important as this work appears conceptually very close.

> On W2-3) I am not convinced a new method is needed here. The propagation is "just" a GNN, and GNNs have been scaled to very large graphs (e.g., by MariusGNN). (Also, it's not clear why this would take 24h on a CPU, as it's cheap per edge operations and edges can be streamed through multiple cores. But that's not the key point.)

Here also the setup is not clear. As backprop is not needed, we can simply stream the edges and aggregate the messages; in parallel over arbitrarily many cores (with the usual tricks).

> On W4-1) DGL-KGE and PyTorch-BigGraph are not SOTA; e.g., MariusGNN is a natural baseline, as are all the newer large-scale GNN works.
> On W4-2) It still needs discussion. And what about TIGER (https://dl.acm.org/doi/10.14778/3675034.3675039)?
See above.

> On W4-3) To see if this is true (it might be), a well-tuned SOTA link prediction method should be trained and compared to. In fact, the response mentions [1], which seems to have done just that (on smaller graphs); so why not compare to SEPAL on these datasets? See also W2-2.

The authors report convincing results here.


Based on the rebuttal, I am now more convinced that the proposed "alignment" is useful in practice, whereas the scalability part appears rather nice to have than a crucial novelty (and, again, it's unclear if BLOCS is needed).

I've upped my rating and downed my confidence to reflect these thoughts.

**Limitations:**

Yes

**Paper Formatting Concerns:**

/

**Quality:**

2

**Strengths And Weaknesses:**

The main strength of this work lies in it's simplicity:

S1. Simple method

S2. Cheap to run

S3. Reasonable heuristic (most frequent entities + neighbors)

The approach as a whole, however, fails to make a convincing case:

W1. Cost arguments not convincing. KGE methods have been scaled to graphs larger than the largest one considered in the experimental study, including many works mentioned in Sec. 2.3 and including works that require few resources. It's not clear if there is a need for another method.

W2. The subgraph propagation part (which is the key technical contribution) is not convincing. This is for three reasons:

W2-1: Analysis limited. For the subgraph propagation step, the authors claim in text that their approach minimizes "a global energy". The actual proposition then limits this to DistMult, identity similarity and "sum of scores" objective; this is does not reflect an actually interesting setting (e.g., sum of scores is not a suitable KGE training objective).

W2-2: The method seems to be equivalent to the one of Albooyeh et al. (EMNLP Findings, 2020, https://aclanthology.org/2020.findings-emnlp.241/).

W3-3: It's not clear why the decomposition method (appendix) is needed for propagation and what its impact is w.r.t. to the analysis of Prop. 4.1. Can't this simply be done on the CPU (e.g., stream edges through)?

W3. Experimental setup not clear. Neither the hyperparameters (which ones there are and which ranges have been explored) nor the search strategy has been described in the paper. This includes hyperparameters of the proposed method as well as the underlying KGE method, and the cost/validation objectives used for the core graph.

W4. Experimental study not insightful. This is for multiple reasons.

W4-1: The paper compares to embeddings trained using PyTorch BigGraph and DGL-KE, neither of which is a state of the art method; i.e., no newer "large KG"-methods are considered.

W4-2: Transductive methods are not timely anymore, esp. in the KG setting, since good inductive methods exist by now; e.g., Cui et al. (NeurIPS 2024, https://proceedings.neurips.cc/paper_files/paper/2024/hash/0d70af566e69f1dfb687791ecf955e28-Abstract-Conference.html). Such methods are neither discussed nor compared to.

W4-3: The proposed method aims to optimize link prediction performance cheaply (as that's the idea behind propagation), and, as expected, falls behind link prediction methods (as shown in the appendix). Nevertheless, the proposed method achieves better vertex classification/regression scores in the experimental study. It's neither explored nor intuitively clear why this is the case.

---

> ### Author Rebuttal · Authors · 2025-07-31
>
> We sincerely thank the reviewer for the detailed review.
> # Preamble
> Before addressing each specific point, we would like to briefly clarify the research problem our paper tackles and the corresponding contributions we aim to make. Understanding this framing helps contextualize the design choices and experimental methodology.
>
> ### **Research problem**
> Our work focuses on extracting knowledge from large-scale knowledge graphs (KGs) through embeddings, with the goal of enriching downstream tabular data to boost performance in supervised learning tasks (e.g., regression and classification). Unlike most prior work, centered on knowledge graph completion (KGC) via link prediction, we target a different application: **downstream feature enrichment using KG embeddings**.
>
> Recent findings [1] have shown that strong performance on KGC does **not** correlate with improved performance on downstream predictive tasks. Despite this, most of the KGE literature continues to prioritize KGC as the sole benchmark, with limited attention to **how the learned embeddings transfer to practical ML tasks beyond the KG itself**.
>
> Our aim is thus not to propose yet another KGC method, but rather to design a method tailored to the demands of downstream prediction tasks. In this context, **scalability is not an end in itself, but a necessary requirement** to unlock the predictive power of large general-purpose KGs in downstream analyses.
>
> [1] Daniel Ruffinelli and Rainer Gemulla. Beyond link prediction: On pre-training knowledge graph embeddings. In Proceedings of the 9th Workshop on Representation Learning for NLP (RepL4NLP-2024), 2024.
>
> ### **Contributions**
> 1. We theoretically argue why **traditional KGE models**, optimized for KGC via local contrastive learning, **are not well-suited for downstream classification/regression tasks** (see lines 266–270).
> 2. We show that **KGC and downstream learning objectives are fundamentally misaligned**: KGC emphasizes **local** discriminative contrasts, while downstream learning benefits from **globally** consistent embeddings across the graph.
> 3. We propose **SEPAL**, a method designed to maximize this global consistency, and we **theoretically analyze** its propagation step to support this claim.
> 4. Through an extensive experimental study across **7 large-scale KGs and 46 downstream tasks**, we demonstrate that **SEPAL outperforms standard KGE methods by a large margin** (on average +30-35% in downstream performance).
> 5. Finally, we show that **SEPAL can scale to ultra-large graphs** (e.g., WikiKG90Mv2 with 91M nodes) using a single 32GB GPU, and that using such large KGs is beneficial for downstream tasks by enabling richer feature enrichment.
>
> # Specific points
> We now address each of the reviewer’s concerns in detail.
>
> ### **W1. “Cost arguments not convincing”; “not clear if there is a need for another method”**
>
> Our goal is not to reduce cost for link prediction, but to introduce a method specifically **optimized for downstream regression/classification**, a setting largely unexplored by existing scalable KGE methods. This justifies the need for another method.
>
> While some prior works (e.g., Marius) scale KGC efficiently, they still rely on **local contrastive training** (with negative sampling), which is ill-suited for learning **globally consistent embeddings**, a key requirement for downstream tasks.
>
> SEPAL offers a new optimization procedure tailored for this setting, and we demonstrate that it **scales easily to ultra-large KGs** (e.g., WikiKG90Mv2) on a single GPU, without complex distributed setups or memory hierarchy tuning, unlike traditional KGE pipelines.
>
> ### **W2-1. Theoretical analysis of the propagation**
> Our theoretical analysis proves that, in the case of DistMult, **SEPAL minimizes a metric of embedding alignment**. Indeed, the proposed energy is a measure of the “amount of alignment” in the embedding space. The equivalence between SEPAL’s propagation and mini-batch projected gradient descent provides the **guarantee of converging to a stationary point of this energy** with embeddings constrained on the unit sphere.
>
> This energy is indeed not a suitable objective for link prediction, but our theoretical analysis on the queriability of embeddings (lines 258-270) indicates that it is a suitable quantity to minimize in order to perform well on downstream regression/classification tasks. Indeed, **downstream learners require globally aligned embeddings** to be able to extract meaningful patterns even between the embeddings of distant nodes. Our experimental results show that SEPAL achieves top performance for downstream tasks, and thus confirm these theoretical insights.
>
> For other base embedding models, our analogy to Arnoldi-like iterations (lines 254-257 and Appendix D.1) can be adapted, and suggests that the embeddings converge toward generalized eigenvectors of an operator that captures both global graph structure and relation embeddings.
>
> ### **W2-2: Comparison to Albooyeh et al.**
> Albooyeh et al. propose an aggregation that is similar to SEPAL’s aggregation during the propagation phase, however the two methods are in no way equivalent, as **they optimize the embeddings in two very distinct ways**:
>
> - **Albooyeh et al.** introduce propagation *within* the standard link prediction pipeline: during training, for each triple $(v, r, u)$, they occasionally (with probability $p$) replace one entity’s embedding with an aggregated version (e.g., $\theta_u \cdot \theta_r$​) before computing the score. However, **training still relies on negative sampling and a classic link prediction loss**, so it continues to optimize for local triple-level contrasts.
>
> - **SEPAL**, in contrast, **explicitly separates the optimization objective**: embeddings for a small core are trained with a classic KGE objective, and then relation-aware propagation is used across the rest of the graph, without negative sampling. This distinction is crucial: **SEPAL removes the need for negative sampling on 95-99% of the graph**, enabling both improved scalability and alignment properties beneficial for downstream regression/classification.
>
> ### **W2-3: Why BLOCS is needed**
> BLOCS enables GPU-based propagation, which is **crucial for scalability**. Without it, propagation must run on CPU and becomes prohibitively slow. To illustrate this, on YAGO4.5, we tried to run CPU-based propagation, which **did not finish after 24 hours**, while with BLOCS, the GPU version **completes in 2.5 hours**.
>
> ### **W3: Clarify experimental setup**
> We thank the reviewer for the opportunity to clarify this point. In the final version, we will include a table detailing the hyperparameter ranges explored for each dataset, along with the number of training configurations tested per KG.
> Regarding the underlying KGE method, we evaluated three standard training objectives: margin ranking loss, binary cross-entropy, and cross-entropy. We found cross-entropy to yield the best performance, and used it consistently for both SEPAL core training and the DistMult baseline in the reported results.
>
> ### **W4-1: Choice of baselines**
> We respectfully emphasize that our goal is not to improve link prediction performance or compete with recent KGE methods on that task. As stated in the preamble, SEPAL is designed specifically for downstream feature enrichment, where the goal is to generate embeddings that boost performance on regression/classification tasks, not link prediction.
>
> **We compare against DGL-KE and PyTorch-BigGraph because they are widely used, scalable KGE systems that represent the standard approach in real-world large-scale settings**. While newer methods exist, they are generally optimized for link prediction and often rely on complex or resource-intensive training schemes (e.g., advanced negative sampling or query embedding). These methods are not optimized for our task and, as prior work [1] shows, strong link prediction does not necessarily translate to strong downstream performance.
>
> ### **W4-2: “no inductive method discussed nor compared to”**
> Our experiments already include **NodePiece**, an inductive method that can scale up to YAGO4.5 (32M entities).
>
> Inductive capabilities can be valuable in continual learning scenarios (i.e., dynamically evolving KGs). In this use case, SEPAL already supports semi-inductive embedding computation (as noted in our response to Reviewer aZkD).
>
> The method cited by the reviewer (Cui et al., NeurIPS 2024) does not scale to the size of the KGs we target, and **to our knowledge, no existing inductive KGE method scales to the size of graphs we consider**. We remain open to including such baselines should they exist.
>
> ### **W4-3: SEPAL performs better on downstream tasks than link prediction**
> We emphasize that the goal of SEPAL is **not** to cheaply optimize link prediction, but to learn embeddings that are better suited for downstream regression/classification (see preamble and response to W2-1). The **propagation step is designed to maximize embedding consistency across the graph, which we argue and show benefits downstream tasks** more than traditional KGE objectives.

---

> > ### Comment · Reviewer_UiQt · 2025-08-01
> > **Response**
> >
> > Thanks for your feedback! I've collected my thoughts briefly below.
> >
> > On W1. My key point is that scalable (and also low-cost / single-GPU) methods exist but have not been considered in the experimental study.
> >
> > On W2-1) This objective is not well-motivated then, and I did not fully understand the alignment discussion in the paper. I found the response somewhat confusing because this objective is not used for the core (why not?), and quality ("alignment") and scalability (core+propagation) arguments are mixed in the paper.
> >
> > On W2-2) A comparison to this work is missing in the paper (both in words and in experimental study).
> >
> > On W2-3) I am not convinced a new method is needed here. The propagation is "just" a GNN, and GNNs have been scaled to very large graphs (e.g., by MariusGNN). (Also, it's not clear why this would take 24h on a CPU, as it's cheap per edge operations and edges can be streamed through multiple cores. But that's not the key point.)
> >
> > On W4-1) DGL-KGE and PyTorch-BigGraph are not SOTA; e.g., MariusGNN is a natural baseline, as are all the newer large-scale GNN works.
> >
> > On W4-2) It still needs discussion. And what about TIGER (https://dl.acm.org/doi/10.14778/3675034.3675039)?
> >
> > On W4-3) To see if this is true (it might be), a well-tuned SOTA link prediction method should be trained and compared to. In fact, the response mentions [1], which seems to have done just that (on smaller graphs); so why not compare to SEPAL on these datasets? See also W2-2.

---

> ### Author Response · Authors · 2025-08-08
> **Final experiments & responses coming shortly**
>
> Thank you for your response!
>
> We are currently finalizing additional experiments that directly address the points you raised, including comparisons to more baselines. The results will be ready and posted in the discussion within the next few hours. We wanted to ensure you are aware, so you have the opportunity to review them before finalizing your evaluation. Thank you for your understanding.

---

> ### Author Response · Authors · 2025-08-09
> **New experiments (part 1)**
>
> Thank you for your response.
>
> ## New experiments
>
> First, we bring more empirical evidence of SEPAL’s advantages over prior methods.
>
> ### **Comparison to SOTA link prediction methods for large-scale KGs**
>
> To strengthen our empirical evaluation, **we added two strong large-scale baselines**:
>
> - GraSH [2], an efficient hyperparameter optimization framework for large-scale KGEs.
>
> - SMORE [3], a scalable KGE method supporting single-GPU training and multi-hop reasoning.
>
> Both were trained with DistMult as the base embedding method, on **Freebase**, using the authors’ released configurations. We report the **normalized $R^2$ scores** on four real-world regression tasks, along with total training time.
>
> | Method  | Housing prices | Movie revenues | US accidents | US elections | Time |
> |---------|:----------------:|:----------------:|:--------------:|:--------------:|:----------:|
> | PBG     | 0.513          | 0.723          | 0.742        | 0.957        | 33h 42m  |
> | DGL-KE  | 0.445          | 0.610          | 0.686        | 0.962        | 10h 50m  |
> | SMORE   | 0.160          | 0.332          | 0.410        | 0.926        | 6h 15m   |
> | GraSH   | 0.601          | 0.862          | 0.810        | 0.961        | 32h 33m  |
> | SEPAL   | **0.868**      | **0.880**      | **0.953**    | **1.000**    | **5h 58m** |
>
> We trained SMORE for 1 million iterations on one GPU, following the authors’ configuration. Its relatively low performance here suggests that longer training could improve results, but **under a 6-hour budget, SEPAL is substantially better**.
>
> GraSH optimizes hyperparameters for link prediction using successive halvings to discard unpromising configurations at low cost. Results show that, after 33 hours of hyperparameter search on one GPU, GraSH produces better embeddings than other baselines. However, **SEPAL remains the best performer on all tasks, and also the fastest method**. We will include these new results in the final version of the paper.
>
> [2] Adrian Kochsiek, Fritz Niesel, and Rainer Gemulla. Start small, think big: On hyperparameter optimization for large-scale knowledge graph embeddings. In Joint European Conference on Machine Learning and Knowledge Discovery in Databases, 2022.
>
> [3] Hongyu Ren, Hanjun Dai, Bo Dai, Xinyun Chen, Denny Zhou, Jure Leskovec, and Dale Schuurmans. Smore: Knowledge graph completion and multi-hop reasoning in massive knowledge graphs. In Proceedings of the 28th ACM SIGKDD Conference on Knowledge Discovery and Data Mining, 2022.
>
> ### **Comparison to Albooyeh et al. (DistMult-ERAvg)**
>
> To show the novelty of our approach compared to Albooyeh et al. we run their DistMult-ERAvg on Mini YAGO3 (129k entities, 1.1M triples). DistMult-ERAvg is not optimized for large graphs and fails with out-of-memory errors on larger graphs.
>
> | Method          | Housing prices | Movie revenues | US accidents | US elections | Time   |
> |-----------------|:----------------:|:----------------:|:--------------:|:--------------:|:--------:|
> | DistMult-ERAvg  | 0.149          | 0.124          | 0.444        | 0.916        | 2h 41m |
> | SEPAL           | **0.276**      | **0.159**      | **0.548**    | **0.929**    | **5m** |
>
> The table above reports performance and runtime on Mini YAGO3, showing that **SEPAL is 32 times faster than DistMult-ERAvg while achieving consistently better scores**. This is not surprising since DistMult-ERAvg follows the classic optimization loop with negative sampling, gradient computations, and parameter updates, and therefore inherits the limitations of traditional KGE methods. The strength of DistMult-ERAvg lies in out-of-sample embedding computation, which SEPAL also supports (see experiment conducted in response to Reviewer aZkD), but with much greater scalability.
>
> If accepted, we will:
> - Add a brief discussion of Albooyeh et al. in Section 2.1 (page 3)
> - Include a detailed comparison of the two methods in Appendix G, together with the empirical results above.

---

> ### Author Response · Authors · 2025-08-09
> **New experiments (part 2)**
>
> ### **Evaluation on prior benchmark (Ruffinelli et al. [1])**
>
> Ruffinelli et al. [1] propose a benchmark for evaluating KGE methods on **downstream classification and regression**. It includes several KGE baselines and the state-of-the-art GNN KE-GCN for entity classification. For each KG (FB15k-237, YAGO3-10, Wikidata5M), they evaluate embeddings on downstream tasks created from entity attributes of the KG.
>
> We originally did not include this benchmark because our goal was to evaluate **external**, real-world tasks (Fig. 2, p. 8) that are independent of a specific KG, to compare the benefits of diverse KGs. In contrast, the Ruffinelli et al. tasks are artificially created and associated with specific KGs that are **orders of magnitude smaller** than those in our main study.
>
> However, we agree that evaluating SEPAL on these datasets complements our main experiments and enables direct comparison with prior work. We used 128-dimensional embeddings to match one of the dimensions in Ruffinelli et al.’s hyperparameter search space. We also used the authors’ released evaluation script for comparable results. In each table, we report:
> 1. The best KGE method for classification from [1]
> 2. The best KGE method for regression from [1]
> 3. KE-GCN results from [1]
> 4. SEPAL results
>
> ### Table 1: Regression results on Wikidata5M (4.8M entities, 21M triples)
> | Method        | Regression RSE (lower is better) |
> |---------------|:----------------------------------:|
> | TransE (STD)  | 0.596                            |
> | SEPAL         | **0.568**                       |
>
> ---
>
> ### Table 2: Classification and regression results on YAGO3-10 (123k entities, 1M triples)
> | Method         | Classification Weighted F1 (higher is better) | Regression RSE (lower is better) |
> |----------------|:-----------------------------------------------:|:----------------------------------:|
> | DistMult (MTT) | 0.746                                         | 0.472                            |
> | TransE (MTT)   | 0.723                                         | 0.441                            |
> | KE-GCN         | 0.700                                         | 0.398                            |
> | SEPAL          | **0.762**                                     | **0.386**                       |
>
> ---
>
> ### Table 3: Classification and regression results on FB15k-237 (14k entities, 272k triples)
> | Method         | Classification Weighted F1 (higher is better) | Regression RSE (lower is better) |
> |----------------|:-----------------------------------------------:|:----------------------------------:|
> | ComplEx (MTT)  | 0.858                                         | **0.394**                            |
> | RotatE (MTT)   | **0.890**                                     | 0.573                            |
> | KE-GCN         | 0.829                                         | 0.501                            |
> | SEPAL          | 0.853                                         | 0.492                       |
>
> The results show that, on YAGO3-10 and Wikidata5M, **SEPAL achieves the best performance for both regression and classification**, consistent with our main results on much larger KGs.
> On FB15k-237, SEPAL has weaker results, but this is not surprising given that **FB15k-237 is very small** (14k entities, 185 to 6500 times smaller than the graphs considered in our paper). For these tiny graphs, which fall outside SEPAL’s intended scope, our method is not adapted because the core becomes too small to learn good representations for the relations and core entities.

---

> ### Author Response · Authors · 2025-08-09
> **Discussion and conclusion**
>
> ## Discussion
> ### **More explanations on propagation**
>
> - *Why not apply this to the core?*
>
> **We need negative sampling for the core training because otherwise the embeddings could collapse** to a single point (e.g., in DistMult, all entity embeddings identical and all relation embeddings as vectors of ones would minimize the objective). However, once we have learned representations for the relations and the core entities, we can use them to propagate without negative sampling: they act as (fixed) **boundary conditions** and prevent collapse in the rest of the graph (lines 241-244).
>
> - *On quality vs. scalability*
>
> Propagation presents both advantages of being good for downstream tasks and computationally efficient. For this reason, in the paper, quality and scalability arguments are sometimes mixed. Section 4, however, is dedicated solely to explaining the quality aspect.
>
> ### **Why we need BLOCS**
>
> Indeed, our propagation algorithm is message-passing-based, like a GNN, but with no learnable weights. BLOCS is specifically tailored for the requirements of our method.
>
> - *Why not apply prior GNN scaling techniques?*
>
> Prior large-scale GNN methods (e.g. MariusGNN) scale by partitioning the graph and processing partitions one by one. Real-world large-scale graphs are power-law graphs, and for these graphs, standard partitioning methods either produce **partitions with multiple connected components** or with no size control (Appendix E.3., Figure 11, page 32).
>
> For classic GNNs, operating on several connected components during training is not a problem: it is still possible to sample neighborhoods for each vertex, and use the nodes’ vector representations to train the GNN.
>
> In SEPAL, however, 95 to 99% of nodes are outer entities with zero-initialized embeddings, and **the only source of signal during propagation is the core**. Therefore, **if a connected component contains only outer entities, no propagation is possible**.
>
> **BLOCS addresses this by producing subgraphs that are connected**, bounded in size (to fit in GPU memory), and well connected to the core (Appendix E.5, Figure 14, page 35).
>
> - *Why is CPU propagation slower?*
>
> On YAGO4.5, CPU propagation took 37 hours vs. 2.5 hours on GPU with BLOCS. Each propagation step processes millions of triples, requiring elementwise multiplication (DistMult) and normalization per embedding. These operations are heavily parallelizable and far faster on GPU than CPU.
>
> ## Conclusion
> We hope this discussion has at least partly **resolved your concerns** regarding our method and **increased your confidence in its originality and ability to perform significantly better than prior work** on downstream tasks.
>
> **We thank you for your thoughtful feedback**, which prompted us to further demonstrate SEPAL’s capabilities in its target domain (downstream regression/classification tasks from large-scale knowledge graphs).

---

### Official Review · Reviewer_da1w · 2025-07-02

**Clarity:** 3
**Significance:** 3
**Originality:** 3
**Rating:** 5
**Confidence:** 4

**Summary:**

SEPAL is an efficient method for large-scale knowledge graph embedding, addressing computational and memory bottlenecks through message passing and the BLOCS algorithm. Its improvements include:Time optimization: Utilizing the message passing mechanism, the embeddings of the core subgraph (selected by degree or a hybrid strategy of highly connected entities) are propagated to the entire graph, preserving the geometric properties of the relationships.Memory optimization: The graph is divided into overlapping subsets via the BLOCS algorithm to accommodate GPU memory constraints. The selection of the core subgraph takes into account both entity density and relationship diversity (such as a hybrid strategy combining node degree and relationship types), ensuring the quality of the embeddings.

**Questions:**

See Weakness.

**Ethical Concerns:**

["NO or VERY MINOR ethics concerns only"]

**Final Justification:**

After the rebuttal and discussion process, I am happy to change my decision to accept this paper.

**Limitations:**

Yes, the proposed method, while promising, reveals certain limitations in the Experiments section.

**Paper Formatting Concerns:**

This paper has no major formatting issues.

**Quality:**

3

**Strengths And Weaknesses:**

Strengths:
1.Significantly improve the performance of downstream tasks

2.A single 32GB V100 GPU can handle graphs with hundreds of millions of nodes (such as the 91 million entities in WikiKG90Mv2), while baseline methods (such as DGL-KE) require distributed computing.

3. It can capture the global structure and enhance the geometric consistency of the embedding by replacing local contrastive learning with relation-aware propagation.

Weaknesses:
1.Experimental design flaw: Insufficient coverage of downstream tasks
Most tasks are based on table regression/classification and have not been validated in complex scenarios such as multi hop inference.
2.In WikiKG90Mv2 link prediction, SEPAL's MRR (0.195) is significantly lower than PBG's (0.632).

---

> ### Author Rebuttal · Authors · 2025-07-31
>
> We thank the reviewer for their thoughtful and positive assessment of our work.
>
> We are glad the review acknowledges the significant improvement in downstream task performance and the ability of SEPAL to scale to massive knowledge graphs using a single 32GB GPU.
>
> Below, we address the reviewer’s comments in more detail.
>
> ### **1. Experimental design and task focus**
> In this work, our primary focus is on **downstream feature enrichment**, a setting where the goal is to learn entity embeddings from a knowledge graph and incorporate them as features into tabular datasets for regression or classification tasks. This setup enables the integration of external knowledge into practical ML pipelines (e.g., incorporating knowledge about cities to better predict housing prices).
>
> This task differs substantially from **link prediction** and **logical query answering**, and requires different optimization objectives. Specifically, while link prediction focuses on distinguishing positive triples from corrupted negatives, encouraging local contrasts, downstream enrichment tasks benefit from global consistency across the graph, so that entities located far apart can still have semantically meaningful and comparable embeddings from which a downstream predictor can learn shared patterns. This distinction is discussed in detail in our analysis of embedding "queriability" (Section 4, page 7), where we provide theoretical justification for why long-range consistency is essential in our setting.
>
> We also argue that SEPAL is well-suited to this objective. In Section 4 (lines 239–257), we provide theoretical arguments showing how our propagation-based approach fosters this global consistency. In contrast, traditional KGE methods rely on negative sampling during training, which increases optimization complexity and can distort long-range consistency by overemphasizing local contrast [1, 2].
>
> To validate these theoretical insights, we conduct a large-scale empirical study spanning **46 downstream tasks** over **7 large-scale KGs**, and show that SEPAL consistently outperforms baselines, achieving **35% higher average performance than PBG** and **30% higher than DGL-KE**.
>
> We agree with the reviewer that **multi-hop inference** is a compelling and valuable research direction. However, we believe that it falls outside the scope of this paper. Such settings typically require **explicit modeling of logical queries during training** and **dedicated architectures for reasoning** [3, 4], which are fundamentally different from our goal of scalable feature enrichment for general-purpose downstream tasks. We chose to focus our contributions on this under-explored setting, and we believe our results demonstrate strong practical relevance and theoretical grounding. Nevertheless, we agree that ideas from SEPAL could be adapted for multi-hop inference in future work.
>
> If the paper is accepted, we will further clarify this distinction in the introduction and conclusion to avoid confusion regarding task scope.
>
> [1] Erik Arakelyan, Pasquale Minervini, Daniel Daza, Michael Cochez, and Isabelle Augenstein. Adapting neural link predictors for data-efficient complex query answering. Advances in Neural Information Processing Systems, 2023.
>
> [2] Pedro Tabacof and Luca Costabello. Probability calibration for knowledge graph embedding models. In International Conference on Learning Representations, 2020.
>
> [3] Hamilton, W., Bajaj, P., Zitnik, M., Jurafsky, D., & Leskovec, J.. Embedding logical queries on knowledge graphs. Advances in Neural Information Processing Systems, 2018.
>
> [4] Ren, H., & Leskovec, J.. Beta embeddings for multi-hop logical reasoning in knowledge graphs. Advances in Neural Information Processing Systems,  2020.
>
> ### **2. Link Prediction Performance on WikiKG90Mv2**
> Following the initial submission, we were able to **substantially improve SEPAL’s MRR from 0.195 to 0.5291**. This improvement was achieved by increasing the number of core training epochs (from 15 to 20), enlarging the batch size (from 8,192 to 16,384), and increasing the number of propagation steps (from 10 to 15). In comparison, PBG achieves an MRR of 0.6325, and DGL-KE achieves 0.3202.
>
> That said, we emphasize that **link prediction is not the target task for which SEPAL was designed** (see Section 4, page 7, lines 266–270). SEPAL’s optimization is geared toward producing globally consistent embeddings that are well-suited for downstream regression and classification tasks, rather than optimizing local triple contrast through negative sampling, as in traditional KGE methods. As a result, as explained above, **SEPAL’s performance for downstream regression and classification tasks is unmatched**.

---

> > ### Comment · Reviewer_da1w · 2025-08-01
> >
> > Thanks for the discussion and it addressed some of my previous concerns.

---

### Official Review · Reviewer_mvhY · 2025-07-02

**Clarity:** 4
**Significance:** 4
**Originality:** 4
**Rating:** 5
**Confidence:** 4

**Summary:**

Knowledge graphs are widely used to store knowledge and can be leveraged to produce embeddings that benefit various applications. However, existing methods usually do not scale well. This paper proposes a method that first obtains embeddings on a small subgraph called core subgraph and then uses message passing to propagate embeddings outside the core subgraph. Extensive experiments corroborate the effectiveness of the proposed method.

**Questions:**

See above.

**Ethical Concerns:**

["NO or VERY MINOR ethics concerns only"]

**Final Justification:**

Both of my concerns are addressed by the authors. The additional experiments based on PageRank scores further  confirm the robustness of the proposed method. I will keep the score unchanged.

**Limitations:**

yes

**Quality:**

3

**Strengths And Weaknesses:**

**Strengths**:

1. The considered problem is of great importance to both academia and industry.

2. The paper is well-written and easy-to-follow.

3. The experimental results are sufficient to prove the effectiveness of the proposed method, from my perspective.

4. The propagation step outside the core subgraph enjoys the advantage of existing knowledge graph embedding models and further affirms their effectiveness.

5. The propagation step is theoretically guaranteed.

**Weaknesses**:

1. The BLOCS algorithm is described in the appendix. The authors could still describe its high-level ideas and main steps in the main paper.

2. The authors mainly determine core subgraphs using node degrees. Other node centrality measures like PageRank score could also be tried.

---

> ### Author Rebuttal · Authors · 2025-07-30
>
> We thank the reviewer for the thoughtful review, insightful suggestions, and positive feedback.
>
> We are glad that the review recognizes the practical significance of the problem we address, the clarity and readability of our paper, and the effectiveness of our method, both empirically and theoretically.
>
> Below, we address the specific comments made by the review.
>
> ### **1. BLOCS in the appendix**
> Due to space constraints, we placed the full description of the BLOCS algorithm in the appendix. However, we agree that including a **high-level overview** in the main paper would improve clarity and accessibility. If the paper is accepted, we will use the additional page to **move the "High-level principle" paragraph from Appendix E.2 (pages 30–31) into the main body**, specifically at the end of Section 3.1, to give readers a clearer understanding of BLOCS without needing to refer to the appendix.
>
> ### **2. Considering other centrality measures for core selection**
> Following the reviewer’s suggestion, **we conducted an additional experiment** where we **replaced degree with PageRank** for core selection (keeping all other hyperparameters unchanged). In our original design, we used node degree as the centrality measure because it is extremely efficient to compute, aligning with our emphasis on scalability. However, **PageRank can also be computed very efficiently** using sparse matrix multiplications.
>
> The table below reports results on the real-world downstream tasks using normalized R² scores and training time (in parentheses).
>
> | Dataset         | Core defined by Degree           | Core defined by PageRank         |
> |-----------------|------------------|------------------|
> | YAGO3           | **0.783** (11m 20s) | 0.742 (9m 23s)   |
> | YAGO4           | 0.817 (6h 20m)     | **0.861** (4h 53m) |
> | YAGO4+T         | 0.815 (10h 9m)     | **0.881** (8h 16m) |
> | YAGO4.5         | **0.949** (4h 11m) | 0.925 (4h 3m)    |
> | YAGO4.5+T       | **0.923** (2h 47m) | 0.912 (2h 40m)   |
> | Freebase        | 0.917 (5h 58m)     | **0.919** (5h 58m) |
> | WikiKG90Mv2     | 0.898 (20h 31m)    | **0.902** (23h 59m) |
>
> **Performance**
>
> Regarding performance, it seems that the difference between degree and PageRank **depends on the graph’s structure**. Specifically, we observe that for YAGO4, YAGO4+T, Freebase, and WikiKG90Mv2, PageRank improves the performance. This echoes the results of Figure 14 (Appendix E.5, page 35), which show that in these four specific KGs, contrary to the others, the degree-based core selection yields cores that are connected through fewer edges to some outer subgraphs. We can therefore conjecture that for these KGs, **PageRank improves information flow and mitigates issues like oversquashing during propagation, ultimately increasing performance**.
>
> **Cost**
>
> Regarding computational cost, we see that **SEPAL with PageRank usually runs slightly faster than with degree**. This is because PageRank yields slightly sparser cores, leading to faster core training.
>
> We will include these findings, along with a brief analysis and discussion, in the final version of the paper.

---

> > ### Comment · Reviewer_mvhY · 2025-08-02
> > **Thanks for the response**
> >
> > Both of my concerns are addressed by the authors. The additional experiments based on PageRank scores further confirm the robustness of the proposed method.
> >
> > Since my evaluation is already positive, I will keep the score unchanged.

---

### Official Review · Reviewer_aZkD · 2025-07-03

**Clarity:** 2
**Significance:** 3
**Originality:** 2
**Rating:** 4
**Confidence:** 3

**Summary:**

The paper proposes two things – an embedding propagation algorithm called SEPAL that scales to large KGs and BLOCS, an algorithm to break large graphs into balanced overlapping subgraphs. The algorithm works in two phases – identifying a core subgraph and creating core and relation embeddings using existing KGE methods and propagating the embeddings to the other subgraphs using message passing. The authors show that this process can be executed in a limited GPU memory setting for really large graphs. The evaluation on downstream applications shows cases the merit of the proposed solution.

**Questions:**

Q1: If the knowledge graph is updated with new entities and relations, does the entire process need to rerun (starting from code graph selection)?
Q2: Is there a dependency on the graph having higher connectivity or connectedness for the SEPAL algorithm?

**Ethical Concerns:**

["NO or VERY MINOR ethics concerns only"]

**Final Justification:**

The authors have mostly addressed the two questions that were raised in the review.

**Limitations:**

Authors have sufficiently addressed the limitations of the work

**Paper Formatting Concerns:**

1. Lines 46 and 108 - What exactly is '...'?
2. Point '1' on line 51 is confusing to read.

**Quality:**

2

**Strengths And Weaknesses:**

S1: The authors have clearly defined the problem and where the proposed solution comes in.
S2: The proposed algorithm SEPAL reuses several methods from the existing graph / KG literature towards scaling of the KGE generation, and the authors also provide theoretical analysis for SEPAL.
S3: The proposed algorithms are evaluated on a broad set of KGs and of varying lengths showcasing the strength of the solution in terms of usage of only commodity hardware to compute the embeddings.
W1: The biggest weakness is that the paper is heavily reliant on the Appendix and while it is appreciated that the authors have provided substantial and important information in the appendix, the main paper should be representative of the core work (both algorithms and the evaluation). In the current format, it takes a significant amount of back and forth to understand the work.
W2: (Minor) The placement of the proposed algorithm BLOCS in appendix is slightly confusing as it is claimed to be a contribution but only mentioned in the main paper.

---

> ### Author Rebuttal · Authors · 2025-07-30
>
> We sincerely thank the reviewer for the insightful feedback.
>
> We are pleased to see the reviewer highlight the clear problem formulation, the comprehensive empirical evaluation across diverse knowledge graphs and downstream datasets showcasing the benefits of SEPAL, and the theoretical analysis supporting our method.
>
> Below, we answer the reviewer’s questions and concerns in detail.
>
> ### **Q1: SEPAL for continual learning**
>
> SEPAL does **not** require rerunning the entire process if the graph is updated with new entities. **New embeddings can be computed via a few additional propagation steps**, as long as the relations remain unchanged.
>
> To demonstrate this, **we conducted an additional experiment** using two versions of the YAGO3 knowledge graph:
> - YAGO3-2014 (original release),
> - YAGO3-2022 (revived version).
>
> The table below gives the statistics of these datasets.
>
> | Dataset       | #Entities   | #Relations | #Triples     |
> |:------------:|:-----------:|:----------:|:------------:|
> | YAGO3-2014    | 2,570,716   | 37         | 5,585,004    |
> | YAGO3-2022    | 4,546,966   | 37         | 14,691,781   |
>
> In the continual learning setting (**SEPAL-CL**), we:
> - Initialized the embeddings of YAGO3-2022 using embeddings precomputed on YAGO3-2014,
> - Propagated embeddings for just 5 steps to update existing nodes and embed new entities.
>
> The table below compares the performance and runtimes of SEPAL-CL, SEPAL trained from scratch, and DistMult trained from scratch. We also include the performance obtained by SEPAL trained on the older YAGO-2014 dataset to show the benefit of updating the knowledge graph.
>
> | YAGO3 version | Method     | Avg. performance (normalized R²) | Runtime       |
> |:-------------:|:-----------:|:---------------------:|:--------------:|
> | 2014          | SEPAL      | 0.836                  | 0h 11m 20s    |
> | 2022          | DistMult   | 0.884                  | 11h 05m 36s   |
> | 2022          | SEPAL      | 0.988                  | 1h 05m 07s    |
> | 2022          | SEPAL-CL   | 0.962                  | 0h 01m 08s    |
>
> Table: Results of the continual-learning experiment, on the real-world downstream tasks.
>
> We see that:
> - For downstream applications, YAGO3-2022 brings value compared to YAGO3-2014 (for all methods, scores are higher than the strongest method on YAGO3-2014, SEPAL).
> - SEPAL from scratch is 10$\times$ faster than DistMult.
> - **SEPAL-CL is 57$\times$ faster than SEPAL from scratch, and 587$\times$ faster than DistMult**.
> - **SEPAL-CL clearly outperforms DistMult**, and almost **matches the performance of SEPAL trained from scratch**, even in this very **challenging scenario** (8 years between the two versions of the graph), where the size of the graph has doubled in terms of entities, and tripled in terms of triples. In a real-life application, we could imagine embeddings recomputed monthly at very low cost.
>
> We sincerely thank the reviewer for this insightful question, which motivated us to conduct this additional experiment. We genuinely believe that **this study increased the impact of our method by demonstrating its applicability to continual learning settings**.
>
> ### **Q2: Dependency on connectedness and connectivity**
> We thank the reviewer for raising this important question. SEPAL requires the input knowledge graph to be **connected**, but this is **not a limiting assumption in practice**: as shown in Table 2 (Appendix A, page 17), the largest connected component in real-world KGs typically accounts for nearly the entire graph.
>
> Regarding graph density, **SEPAL can handle a wide range**. Table 2 (Appendix A, page 17) shows that we applied SEPAL to KGs with densities spanning several orders of magnitude (from $6 \times 10^{-8}$ to $1 \times 10^{-4}$). Across this spectrum, SEPAL consistently achieved the best performance on downstream tasks.
>
> The **primary impact of higher density is computational**: denser graphs lead to denser core subgraphs, which can increase the time required for the core training phase. In such cases, it can be beneficial to slightly reduce the size of the core subgraph to maintain efficiency without significantly affecting performance.
>
> ### **W1-2: Reliance on the Appendix**
> Due to the strict 9-page limit, we moved detailed descriptions and ablations to the appendix to keep the main narrative focused and accessible. That said, we fully agree that improved self-containment in the main paper would aid readability.
>
> Therefore, if the paper is accepted, **we will use the additional page allowed to move the paragraph “High-level principle” of Appendix E.2 (pages 30-31), which describes BLOCS, to the end of section 3.1 in the final version of our paper**. We will also summarize extra evaluation results in the main text to minimize back-and-forth. We believe these changes will significantly improve the clarity and accessibility of our contributions.
>
> ### **Paper formatting**
> We will address the paper formatting concerns raised by the review in the final version of the paper, and thank the reviewer for pointing them out.

---

> > ### Comment · Reviewer_aZkD · 2025-08-01
> >
> > Thank you authors for addressing my questions and concerns around the weaknesses. While the authors have provided additional experimentation results when entities are added, it only partially addresses the question as the impact of adding relations was not addressed. I have updated my score to a 4 as I am still skeptical if the said update to the paper post accept can address the reliance on the appendix.

---

### Note · Authors · 2025-08-12

We thank all reviewers for their constructive feedback, which led us to clarify our contributions and run additional experiments that further confirm SEPAL’s effectiveness.

We would like to **provide more context regarding our latest response to Reviewer UiQt, which unfortunately was sent after their final rating**.

Reviewer UiQt expressed concerns regarding our initial baselines (PyTorch-BigGraph, DGL-KE), arguing that newer methods (e.g., Marius) or well-tuned state-of-the-art link prediction models should be included to fairly assess SEPAL’s downstream performance on large-scale KGs.

Our response addressed these concerns by:

### **1. Adding newer baselines**

- We added **SMORE** [3], a newer large-scale KGE method with training times similar to Marius (Table 5 in [3]). On Freebase, SMORE trains in ~6h (same as SEPAL) but delivers significantly lower downstream performance. This showcases **SEPAL’s competitive training times**.
- To compare against a **well-tuned large-scale baseline, we also added GraSH** [2], a hyperparameter optimization framework for large-scale KGEs, that selects configurations that perform the best on link prediction. Results show that **SEPAL still achieves higher downstream scores**.

Note: We attempted to include Marius/MariusGNN, but installation was unsuccessful despite repeated efforts. However, we point out that **Marius does not increase link prediction performance compared to DGL-KE and PyTorch-BigGraph**, but rather decreases the time needed to achieve this performance, as shown in the Marius paper itself (https://arxiv.org/pdf/2101.08358, Tables 2-3-4-5).

### **2. Evaluating SEPAL on a prior benchmark with smaller datasets**

As suggested, we evaluated SEPAL on smaller KGs from the Ruffinelli et al. [1] benchmark, which **includes multiple tuned KGE baselines and a GNN SOTA for entity classification** (KE-GCN [4]). On YAGO3-10 and Wikidata5M, **SEPAL achieves the best performance for both regression and classification**, confirming our main results on much larger KGs. Lower performance on FB15k-237 is due to its tiny size (the core subgraph becomes too small to learn good embeddings for relations and core entities), which is outside SEPAL’s target domain.

[1] D. Ruffinelli & R. Gemulla. Beyond link prediction… RepL4NLP, 2024.

[2] A. Kochsiek et al. Start small, think big… ECML PKDD, 2022.

[3] H. Ren et al. Smore… KDD, 2022.

[4] D. Yu et al. Knowledge embedding based graph convolutional network. WWW, 2021.

---

### Decision · Program_Chairs · 2025-09-17

**Decision:**

Accept (poster)

**Comment:**

This paper proposes SEPAL, an approach to learn knowledge graph embeddings on large graphs which can be used to enrich downtream task applications like classification and regression.  The authors propose a two stage approach which involves computing embeddings over a small subgraph (core subgraph) and propagating these over the rest of the graph.  The authors discuss optimizations for training as well as show improvements in performance of embeddings learned with SEPAL on downstream tasks.

This paper leans towards acceptance, and the rebuttal efforts were generally appreciated by reviewers, some of whom raised scores.  Authors introduced new baselines and some referenced benchmarks as well, which do not detract from their claims.

Main feedback points:

- this paper is heavily reliant on appendix and the main content of the paper doesn't stand well on its own without frequent referencing to the appendix.  This brings down the evaluation and I encourage the authors to find ways to better focus on the key points to communicate in the main paper (aZkD, mvhY)

- some reviewers took issue with baseline choice (e.g. adopting newer baselines like Marius or TIGER), as well as inclusion of other downstream tasks (UiQt)

- there were some questions about theoretical and conceptual clarity (rationale behind the chosen core selection method, theoretical analysis of global energy objective, and relative novelty of the propagation step compared to prior works (UiQt, mvhY)